# Can Hamilton's rule be violated?

Matthijs van Veelen*

CREED, University of Amsterdam, Amsterdam, The Netherlands

**Abstract** How generally Hamilton's rule holds is a much debated question. The answer to that question depends on how costs and benefits are defined. When using the regression method to define costs and benefits, there is no scope for violations of Hamilton's rule. We introduce a general model for assortative group compositions to show that, when using the counterfactual method for computing costs and benefits, there is room for violations. The model also shows that there are limitations to observing violations in equilibrium, as the discrepancies between Hamilton's rule and the direction of selection may imply that selection will take the population out of the region of disagreement, precluding observations of violations in equilibrium. Given what it takes to create a violation, empirical tests of Hamilton's rule, both in and out of equilibrium, require the use of statistical models that allow for identifying non-linearities in the fitness function.
DOI: https://doi.org/10.7554/eLife.41901.001

## Introduction

Hamilton's rule (*Hamilton, 1964a*; *Hamilton, 1964b*) states that pro-social, altruistic behaviour will be selected for if $rb>c$, where $b$ are the benefits to the recipient, $c$ the costs to the donor, and $r$ is the relatedness between them. There is however no consensus concerning how generally this rule applies. Some claim that Hamilton's rule is completely general (*Abbot et al., 2011*). Others claim that is almost always wrong (*Nowak et al., 2010*). Recently it has been suggested that the reason why there is disagreement about the generality of Hamilton's rule, is that different participants in the debate have different definitions of costs and benefits (*Birch, 2014*; *Birch and Okasha, 2015*; *van Veelen et al., 2017*). Some define costs and benefits using the *regression method* (*Gardner et al., 2011*; *Marshall, 2011*). With this definition, Hamilton's rule is claimed to always hold, and this version of Hamilton's rule therefore is also referred to as the general version of Hamilton's rule, or HRG (*Birch, 2014*). Others use the *counterfactual method* to determine costs and benefits (*Karlin and Matessi, 1983*; *Matessi and Karlin, 1984*; *Matessi and Karlin, 1986*; *van Veelen et al., 2017*). With this definition, Hamilton's rule is claimed to hold only if the interaction is characterized by 'generalized equal gains from switching', or, in other words, if the fitness effects of one individual changing from defection to cooperation are independent of the behaviour of the others, including the recipient (*van Veelen et al., 2017*).

In what follows, we will consider both definitions, and explore the scope for violations. With the regression method, we will see that there is an identification problem; there are cases in which there are actually multiple linear specifications, leading to multiple Hamilton's rules, all of which hold. Hamilton's rule, using the regression method, therefore is not necessarily uniquely defined. For every given specification, however, Hamilton's rule cannot be violated. This is a short summary of a point made in Section 4 of *van Veelen et al., 2017*.

In the main part of the paper, we will present a general model of assortative group compositions, which we combine with the counterfactual method for computing costs and benefits. For this model, we define *population structure profiles*, which reflect the distribution of group compositions that the population structure puts typical mutants in. In combination with the shape of the fitness function, these population structure profiles determine whether invading co-operators, or invading defectors, have a selective advantage. We will find that this allows for violations of Hamilton's rule. Part of the

*For correspondence:
c.m.vanveelen@uva.nl

Competing interests: The author declares that no competing interests exist.

reason why these violations can occur, is that relatedness $r$ in Hamilton's rule is a one-dimensional measure for population structure, while the population structure profile that is relevant for the direction of selection is a richer description of population structure. Violations in equilibrium, however, may not always be possible, because selection can also take a population out of the region where Hamilton's rule and the direction of selection disagree. This happens, for instance, in examples with synergies.

In the final section, we discuss implications for empirical tests of Hamilton's rule. The empirical literature is regularly lacking a precise description of what violations would look like, or how to identify them in the data. Because the regression method does not allow for violations, an empirical test of Hamilton's rule, using the regression method, is not a meaningful exercise. With the counterfactual method, violations are possible, although observing them either requires studying out-of-equilibrium dynamics, or studying systems that allow for in-equilibrium violations. In both cases non-linear statistical models should be allowed for.

## The regression method

The regression method defines costs and benefits according to an ordinary least squares regression. The fitness of individuals is regressed on two or more variables. One of those variables is their own level of cooperation, which may be a binary value, in case there are only co-operators and defectors. We will denote this variable by $x_{se}$ – with $se$ for self – and minus the regression coefficient of that variable is then defined as the cost of cooperation. The other variables are levels of cooperation for different types of interactants – such as, for instance, $x_{si}$ for siblings and $x_{co}$ for cousins. The benefits of having a cooperative sibling then is the regression coefficient of $x_{si}$, and the benefits of having a cooperative cousin is the regression coefficient of $x_{co}$. Many models and many empirical studies focus on one type of interaction – such as interactions between siblings only – but here we also want to discuss the issue of model specification, and therefore it will be useful to allow for the possibility that there are different types of interactions happening at the same time – as Hamilton did in the original paper (*Hamilton, 1964a*; *Hamilton, 1964b*). This allows us to consider different specifications. In what follows, we will think of an example in which both siblings and cousins may have an effect, where roman numeral I refers to a linear specification that includes siblings only, and II to a linear specification that includes both siblings and cousins.

The relatedness between siblings in this version of Hamilton's rule is the covariance of $x_{se}$ and $x_{si}$, divided by the variance of $x_{se}$, and the relatedness between cousins is defined in the same way. With this definition of costs, benefits and relatedness, Hamilton's rule always holds, but it is important to realize that it does so regardless of the linear specification that is chosen (*van Veelen et al., 2017*). In our example, with siblings and cousins, that means that for any two time periods, be it in a model or in a dataset, the change in average cooperativeness $\Delta \bar{x}$ equals both

$$\Delta \bar{x} = r_{si} b_{si,\mathrm{I}} - c_{\mathrm{I}}$$

and

$$\Delta \bar{x} = r_{si} b_{si,\mathrm{II}} + r_{co} b_{co,\mathrm{II}} - c_{\mathrm{II}}$$

The costs and benefits of cooperation carry and index I or II in the subscript, because the value of the regression coefficients may depend on the specification (see *Figure 1*). Not including cousins in the specification means that $b_{co,\mathrm{I}}$ is set to 0. It is important to realize that Hamilton's rule holding does not mean that the specification chosen accurately reflects the way in which fitness's depend on whether one is a co-operator oneself, and on how many siblings and how many cousins are co-operators. Neither specification might represent the true fitness function, and still Hamilton's rule will hold for both. Relatednesses here do not depend on the model specification (see Section 4 of *van Veelen et al., 2017* for a formal derivation).

Hamilton's rule according to the regression method therefore is not necessarily uniquely defined; only if it happens to be the case that $b_{co,\mathrm{II}} = 0$ do both versions coincide in this example. To make it uniquely defined, the regression method would need to be combined with a way to choose between specifications – and with data, applying statistical tests seems to be a natural way to do that. Such a way to choose should then also be applied, not just when choosing between linear specifications,

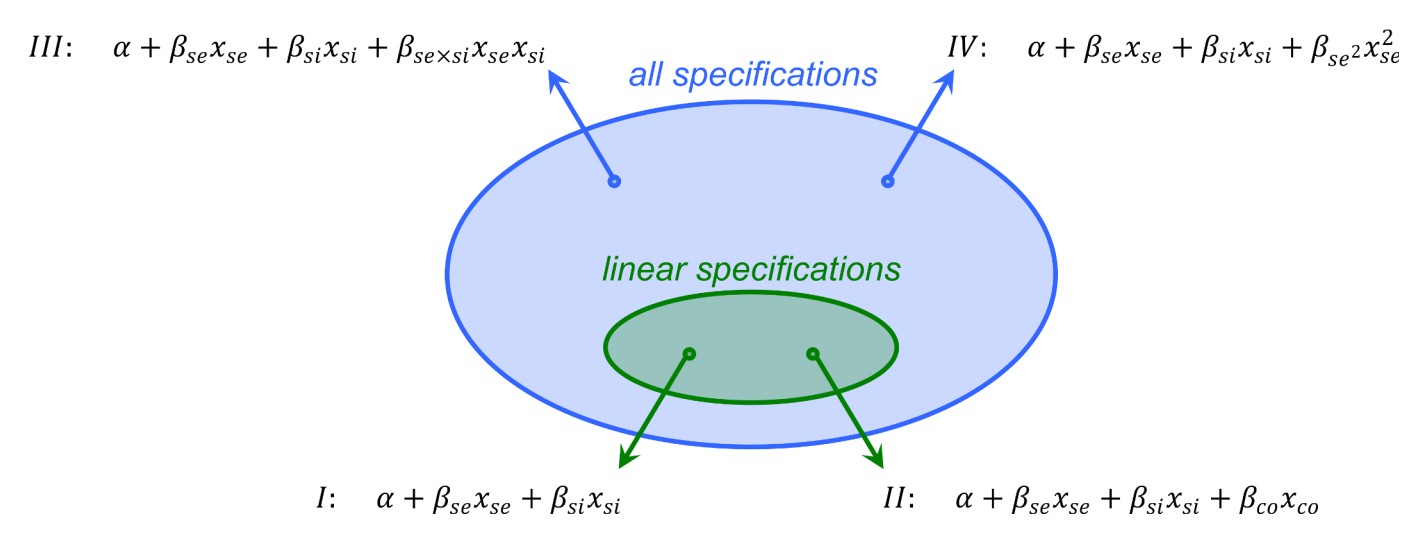

**Figure 1.** The value of coefficients $\beta_{se}$ and $\beta_{si}$ may depend on the specification chosen. If $x_{co}$ is included (as in specification II), these values will be different from when $x_{co}$ is not included (specification I). Including an interaction term (III) or a quadratic term (IV) will also make a difference for the value of $\beta_{se}$ and $\beta_{si}$. All specifications that are linear, result in Hamilton's rules, all of which agree with the direction of selection. Hamilton's rule with specification I says that $r_{si}b_{si,I} - c_I > 0$ if and only if $\Delta\bar{x} > 0$ – where $b_{si,I}$ is the value of $\beta_{si}$, and $c_I$ is minus the value of $\beta_{se}$ in this specification. Hamilton's rule with specification II says that $r_{si}b_{si,II} + r_{co}b_{co,II} - c_{II} > 0$ if and only if $\Delta\bar{x} > 0$ – where $b_{si,II}$ is the value of $\beta_{si}$, $b_{co,II}$ is the value of $\beta_{co}$, and $c_{II}$ is minus the value of $\beta_{se}$ in this specification.
DOI: https://doi.org/10.7554/eLife.41901.002

but also when choosing between linear and non-linear ones, and between different non-linear ones. Hamilton's rule according to the regression method therefore cannot both be uniquely defined, and fully general, because having it well-defined would imply that in some cases non-linear specifications would have to be chosen, while Hamilton's rule being general crucially depends on the specification being linear. But even if Hamilton's rule is not always uniquely defined, it still holds for any given linear specification. Whichever linear specification is chosen, this should therefore never lead to a violation.

## The counterfactual method, applied to a general model of assortment in groups of equal size

With the counterfactual method, the cost of cooperation is defined as the difference between an individual's fitness if it defects, and its fitness if it cooperates. Similarly, the benefits to another individual are defined as the difference cooperation makes for that other individual's fitness. In the relatively simple setup considered here, we assume that individuals interact within groups of size $n$, which implies that a model of dyadic interactions would mean $n = 2$. Their fitnesses will depend on the number of co-operators in their interaction group, and on whether they are a co-operator or a defector themselves; $\pi_C(i)$ is the fitness of a co-operator in a group that contains $i$ co-operators, including the individual itself, and $\pi_D(i)$ is the fitness of a defector in a group that contains $i$ co-operators. The cost of cooperation an individual faces, as well as the benefits cooperation confers on the others, thereby may also depend on what the rest of the group it finds itself in consists of.

A population structure here is a function that represents how the composition of the population depends on the overall frequency $p$ of co-operators; $f_i(p)$ is the fraction of groups with $i$ co-operators in it, at overall frequency $p$. These have to be defined consistently, so that these frequencies always add up to one ($\sum_{i=0}^{n} f_i(p) = 1$), and so that $p$ is indeed the overall frequency of co-operators ($\sum_{i=0}^{n} \frac{i}{n} f_i(p) = p$). The cost of cooperation in a population is now the average cost, given the distribution of group types, and these may very well vary with $p$.

The counterfactual method dates back to *Karlin and Matessi, 1983*, *Matessi and Karlin, 1984*, and *Matessi and Karlin, 1986*, who in their evaluation of whether or not Hamilton's rule holds

moreover did not allow for costs to vary with $p$. We do allow for costs and benefits to depend on $p$. Section 3 in *van Veelen et al., 2017* also discusses some additional differences between the original definition and the one used here.

## Selection

In order to answer the question whether or not mutant co-operators will be able to invade a population of defectors, one would have to consider the type of group the average mutant co-operator would find itself in, given the population structure. The probability that a mutant co-operator finds itself in a group with in total $i$ co-operators is

$$\underline{u}_i = \lim_{p\downarrow 0}\frac{\frac{i}{n}f_i(p)}{p}, \quad i=0,...,n$$

The vector $\underline{\mathbf{u}}$, with elements as defined above, could be called the "population structure profile" at $p=0$. Its counterpart $\overline{\mathbf{u}}$ at $p=1$ will be defined in the same way:

$$\overline{u}_i = \lim_{p\uparrow 1}\frac{\frac{n-i}{n}f_i(p)}{1-p}, \quad i=0,...,n$$

Co-operators can invade defectors if their average fitness at $p=0$ is larger than the average fitness of defectors, who at $p=0$ only find themselves in groups with defectors only, for any population structure.

$$\sum_{i=1}^{n}\underline{u}_i\pi_C(i)>\pi_D(0) \tag{1}$$

Defectors on the other hand can invade co-operators if their average fitness at $p=1$ is larger than the average fitness of co-operators, who at $p=1$ only find themselves in groups with co-operators only.

$$\sum_{i=0}^{n-1}\overline{u}_i\pi_D(i)>\pi_C(n) \tag{2}$$

## Hamilton's rule

Hamilton's rule says that cooperation will be selected for if $rb>c$. We will rewrite this, at $p=0$ and at $p=1$, so as to get inequalities that look more like *Equations 1 and 2*. Relatedness $r$ measures how much more likely co-operators are to be matched with other co-operators, compared to how likely defectors are to be matched with co-operators; $r = P(C|C) - P(C|D)$. In the limit of $p\downarrow 0$, the share of defectors that is matched with co-operators goes to 0, whatever the population structure. With population structure, the average mutant co-operator however might encounter fellow mutant co-operators. This therefore reduces to $r = P(C|C)$, which one can rewrite as $r = \sum_{i=0}^{n-1}\underline{u}_i\frac{i-1}{n-1}$. With the counterfactual method, aggregate benefits at $p=0$ are $(n-1)[\pi_D(1)-\pi_D(0)]$, while costs are $\pi_D(0)-\pi_C(1)$. With those, one can rewrite Hamilton's rule at $p=0$ as follows (see Appendix 1 for more details):

$$\sum_{i=1}^{n}\underline{u}_i\{\pi_C(1)+(i-1)[\pi_D(1)-\pi_D(0)]\}>\pi_D(0) \tag{3}$$

Comparing the equation that indicates when cooperation is selected for (*Equation 1*) and the one for Hamilton's rule (*Equation 3*), we first of all find that they are one and the same equation if the fitness function satisfies $\pi_C(i)=\pi_C(1)+(i-1)[\pi_D(1)-\pi_D(0)]$. In this case, every additional co-operator increases the payoff of a fellow co-operator by just as much as the first co-operator increased the payoff of the defectors in an otherwise all-defector group. Hamilton's rule will then hold at $p=0$ for any population structure profile $\underline{\mathbf{u}}$.

If the fitness function satisfies $\pi_C(i) \geq \pi_C(1)+(i-1)[\pi_D(1)-\pi_D(0)]$ for all $i=1,...,n$, then Hamilton's rule can be violated at $p=0$. Many cases with synergies would fall under this category. In this case Hamilton's rule can, however, still not be violated by a population that is in equilibrium at $p=0$. If cooperation is selected against at $p=0$, then that by definition means that *Equation 1* does

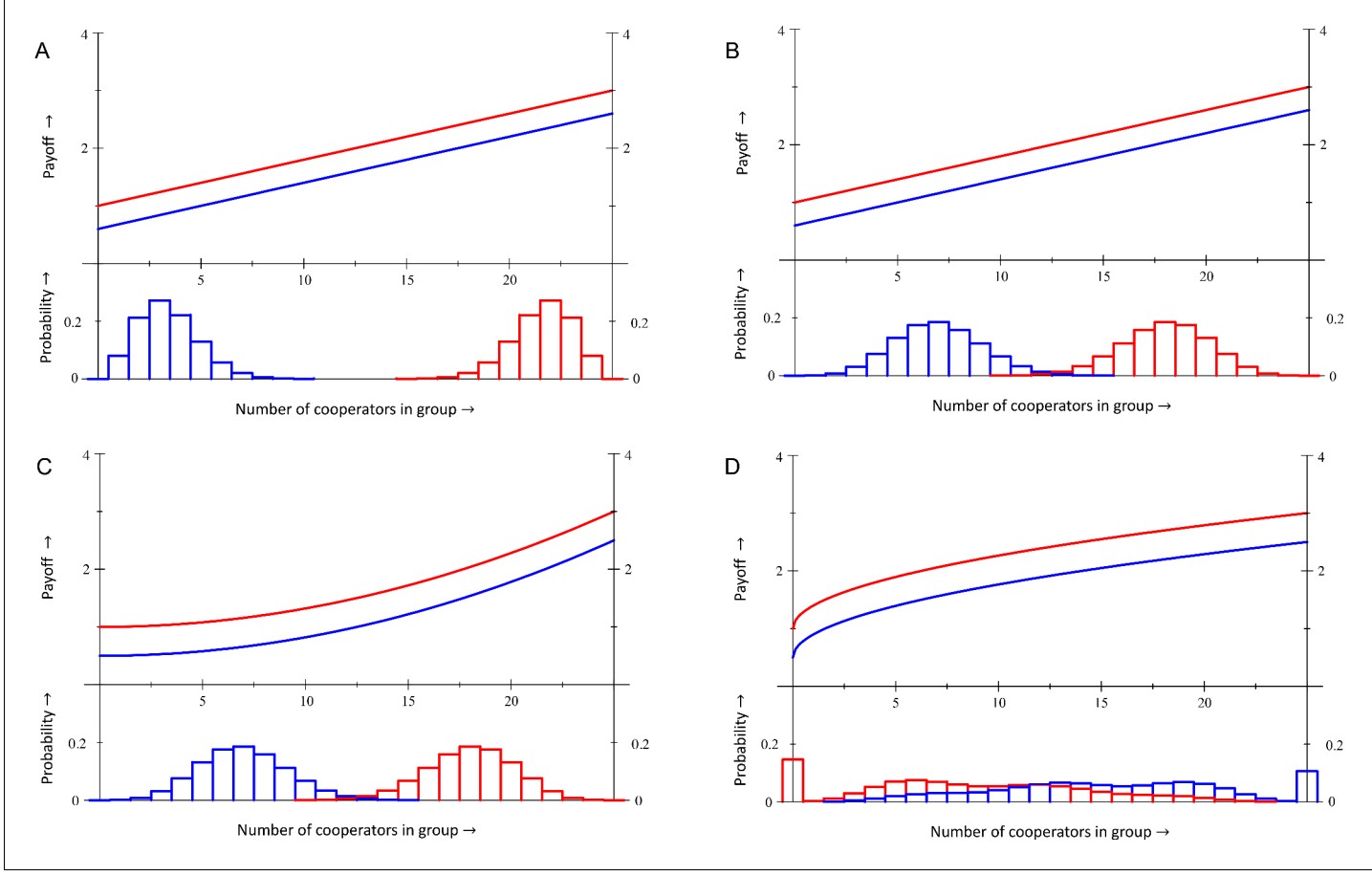

**Figure 2.** Within each panel, the fitness functions are depicted in the upper part. In panels A, B and C the bottom part depicts population structure profiles of mutant co-operators at $p = 0$ (blue) and of mutant defectors at $p = 1$ (red). In panel D the bottom part depicts the distribution of what group types co-operators (blue) and defectors (red) find themselves in, both at the same intermediate equilibrium value for $p$. **No violations of Hamilton's rule with equal gains from switching**. In panels A and B, the fitness function is $\pi_C(i) = 0.6 + 2(i/n)$ and $\pi_D(i) = 1 + 2(i/n)$. In panel A the difference in average fitness between co-operators and defectors is $\overline{\pi}_C - \overline{\pi}_D = -0.128$, both at $p = 0$ and at $p = 1$. Cooperation therefore is selected against at both ends. Inclusive fitness is also $-0.128$ at both ends. Panel B has a more assorted population structure, for which this difference, as well as inclusive fitness, is $+0.16$ at both ends, and cooperation is selected for. **No violations in equilibrium with synergies**. Panel C has the same population structure profiles as panel B, but a different fitness function: $\pi_C(i) = 0.5 + 2(i/n)^2$ and $\pi_D(i) = 1 + 2(i/n)^2$. Here cooperation is selected against at $p = 0$, where $\overline{\pi}_C - \overline{\pi}_D = -0.33$, and selected for at $p = 1$, where $\overline{\pi}_C - \overline{\pi}_D = +0.45$. Inclusive fitness is $-0.48$ at $p = 0$ and $+0.6$ at $p = 1$. **Violation in a mixed equilibrium**. In panel D, the fitness function is $\pi_C(i) = 0.5 + 2(i/n)^{0.5}$ and $\pi_D(i) = 1 + 2(i/n)^{0.5}$. Here, $\overline{\pi}_C - \overline{\pi}_D = 0$ at $p = 0.473$ – which makes it an equilibrium – while inclusive fitness is $0.113 \neq 0$. Details are in Appendix 1, as are computations of inclusive fitness with costs and benefits according to the regression method instead of the counterfactual method.

DOI: https://doi.org/10.7554/eLife.41901.003

not hold. The left hand side of *Equation 3* now is even smaller than the left hand side of *Equation 1*, so if *Equation 1* does not hold, and co-operators are selected against at $p = 0$, also *Equation 3* does not hold, and $rb<c$. This implies that there is no violation. If, on the other hand, cooperation is selected for at $p = 0$, then, by definition, *Equation 1* does hold. Now it is possible that *Equation 3* does not hold, and $rb<0$, but since cooperation is selected for, the population moves away from $p = 0$, and therefore it moves away from where the violation is. Therefore the violation at $p = 0$ cannot be observed in equilibrium.

At the other end, where $p = 1$, Hamilton's rule can be rewritten in a similar way. Here, $rb<c$ if

$$\sum_{i=0}^{n-1} \overline{u}_i \{\pi_D(n-1) - (n-i-1)[\pi_C(n) - \pi_C(n-1)]\} > \pi_C(n) \tag{4}$$

Here, *Equation 2* and *Equation 4* are one and the same equation if the fitness function satisfies $\pi_D(i) = \pi_D(n-1) - (n-i-1)[\pi_C(n) - \pi_C(n-1)]$. In this case, every additional defector decreases the payoff of a fellow defector by just as much as the first defector did to the co-operators in an otherwise all-co-operator group. If this is true, then Hamilton's rule will hold at $p=1$ for any population structure profile $\bar{\mathbf{u}}$.

If the fitness function satisfies $\pi_D(i) \geq \pi_D(n-1) - (n-i-1)[\pi_C(n) - \pi_C(n-1)]$ for all $i = 0, ..., n-1$, then Hamilton's rule can be violated at $p=1$, but, again, not in equilibrium. If defection is selected against at $p=1$, then *Equation 2* by definition does not hold. Under the condition on the fitness function, the left hand side of *Equation 4* is now even smaller than the left hand side of *Equation 2*, so if defectors are selected against at $p=1$, then also *Equation 4* does not hold, and $rb > c$. This, again, implies that there is no violation. If, on the other hand, defection is selected for at $p=1$, there can be a violation, but this cannot be observed in equilibrium, because defectors being selected for means that the population actually moves away from where the violation is.

So far we have an inequality that prevents violations in equilibrium at $p=0$, and another one that prevents violations at $p=1$. Both concern only the fitness function. If, on top of that, the population structure and the fitness function combined imply that the difference in fitness between co-operators and defectors increases in $p$, then the system does not allow for any violations in equilibrium. The reason is that the latter condition would imply that there simply are no stable interior equilibria. Many systems with synergies will satisfy all three conditions, and therefore preclude in-equilibrium violations of Hamilton's rule. Systems with anti-synergies are much more conducive to violations in equilibrium, especially in mixed equilibria, in which co-operators and defectors coexist. Both *Figures 2* and *3* illustrate that.

The fitness function used in panels A and B of *Figure 2* is linear in the number of co-operators in the group. This makes *Equation 1* and *Equation 3* coincide, as well as *Equation 2* and *Equation 4*. With the quadratic fitness function in panel C, both $\pi_C(i) \geq \pi_C(1) + (i-1)[\pi_D(1) - \pi_D(0)]$ and $\pi_D(i) \geq \pi_D(n-1) - (n-i-1)[\pi_C(n) - \pi_C(n-1)]$ hold, which implies that there can be no violation in equilibrium, either at $p=0$ or at $p=1$. This fitness function would be unambiguously synergistic; benefits from cooperation increase, and costs decrease with $i$. Combined with a simple population structure, elaborated on in Appendix 1, $\overline{\pi}_C(p) - \overline{\pi}_D(p)$ moreover increases in $p$, precluding any in-equilibrium violation. Panel D gives an example of a fitness function with anti-synergies, where cooperation is selected for at $p=0$, and selected against at $p=1$, and where there is an equilibrium frequency of co-operators in between $0$ and $1$. At that equilibrium $p$, inclusive fitness is not $0$, which makes Hamilton's rule disagree with the direction of selection.

Games with 2 players and 2 actions are naturally subsumed under this framework, and they allow for whole trajectories to be depicted (*Figure 3*). A natural way to define population structures with 2 players and constant relatedness $r$ would be to choose $f_0(p) = (1-r)(1-p)^2 + r(1-p)$; $f_1(p) = (1-r)2p(1-p)$; and $f_2(p) = (1-r)p^2 + rp$. In this case $\underline{\mathbf{u}} = [0, 1-r, r]$ – implying that a mutant co-operator never finds itself in a group with 2 defectors, obviously; faces a defector with probability $1-r$; and another co-operator with probability $r$ – and $\overline{\mathbf{u}} = [r, 1-r, 0]$. This is combined with two prisoners dilemma's; one with synergies for panel A:

|   | C | D |
|---|---|---|
| C | 3 | 0.1 |
| D | 3.1 | 2 |

and one with the opposite for panel B:

|   | C | D |
|---|---|---|
| C | 3 | 1.9 |
| D | 4.9 | 2 |

In the first one, selection always takes the population out of the region where inclusive fitness disagrees with the direction of selection. In the second one there are mixed equilibria, where neither co-operators or defectors are selected for, while inclusive fitness is not $0$.

A similar point is made with Figures 30 and 31 in *van Veelen et al. (2017)*. The conclusion there – that there is 'no scope for finding violations in equilibria where either one has gone to fixation' – is too strong though; with anti-synergies, violations at $p=0$ and $p=1$ are possible.

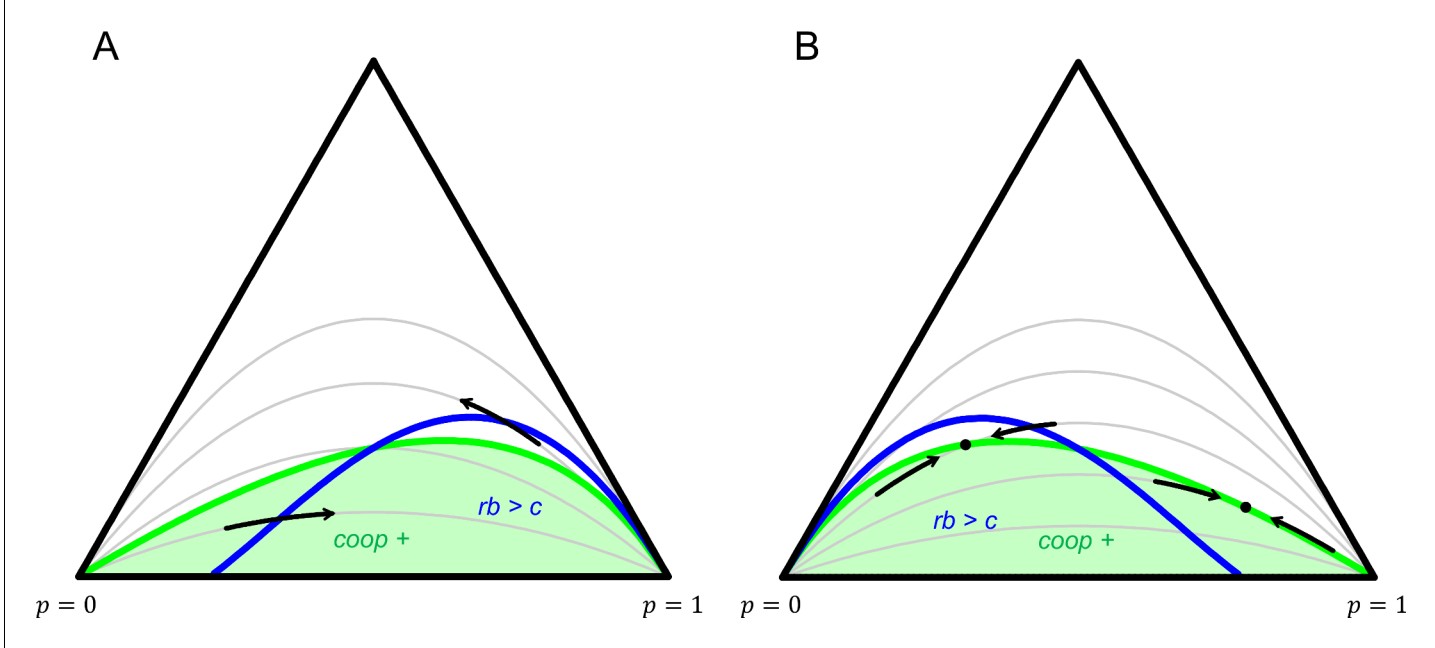

**Figure 3.** Dynamics for two 2-player games. Every point in the simplex represents a population state $(f_0, f_1, f_2)$. The left down corner is $(1, 0, 0)$, which has only groups with 0 co-operators; the right down corner is $(0, 0, 1)$, which has only groups with 2 co-operators; the top corner is $(0, 1, 0)$, which has only groups with 1 co-operator. The grey lines represent different population structures, all with constant relatedness. Any given grey line gives a population state for every overall frequency $p$ of co-operators. Dynamics make populations move along the line that represents the population structure it faces. All grey lines go through the left down corner, where $p = 0$, and the right down corner, where $p = 1$. The straight line on the bottom reflects a totally assorted population that has no mixed groups. The higher up, the more mixed groups there are, and the less assortment there is. The highest up grey line represents a well-mixed population. **No violations of Hamilton's rule in equilibrium with synergies**. In panel A, $\pi_C(1) = 0.1$, $\pi_C(2) = 3$, $\pi_D(0) = 2$ and $\pi_D(1) = 3.1$. The regions where cooperation is selected for (green), and where inclusive fitness is positive (blue) are not the same, but selection always takes populations out of the parts where they disagree. **Violations with anti-synergies**. In panel B $\pi_C(1) = 1.9$, $\pi_C(2) = 3$, $\pi_D(0) = 2$ and $\pi_D(1) = 4.9$. Here populations can settle at mixed equilibria, while inclusive fitness is not 0. Violations at $p = 0$ and $p = 1$ are also possible for more extreme choices of $r$.

DOI: https://doi.org/10.7554/eLife.41901.004

Whether or not Hamilton's rule will hold, with costs and benefits defined according to the counterfactual method, depends on the combination of fitness function and population structure. How often we should expect Hamilton's rule to hold therefore depends on what we think the distribution of fitness functions and population structures is. Whether reasonable assumptions concerning these distributions lead to many or not so many violations of Hamilton's rule is an interesting question for further research that we do not pursue here.

## Implications for empirical tests of Hamilton's rule

There is a number of studies that explicitly set out to test Hamilton's rule empirically. A recent survey by *Bourke (2014)* includes *Bourke (1997)*; *Emlen and Wrege (1989)*; *Gadagkar (2010)*; *Hogendoorn and Leys (1993)*; *Krakauer (2005)*; *Loeb (2003)*; *Metcalf and Whitt (1977)*; *Nonacs and Reeve (1995)*; *Noonan (1981)*; *Pfennig et al. (1999)*; *Queller and Strassmann (1988)*; *Richards et al. (2005)* and *Stark (1992)*. These studies typically consider a behaviour that is present, and therefore presumably selected for, estimate its benefits and costs by linear regression, estimate relatedness, and decide that Hamilton's rule holds if $rb > c$, and is violated if $rb < 0$, where the $r$, $b$ and $c$ now refer to the estimated values of relatedness, costs and benefits. Although intuitively appealing, it is worth realizing that this does not constitute a test of either of the two versions of Hamilton's rule – where it should be noted that these empirical studies were done before it was even recognized that there are different ways to define costs and benefits to begin with.

If we think of Hamilton's rule with costs and benefits defined according to the regression method, then we have seen that this version of Hamilton's rule will hold for any linear specification, applied to any thinkable way in which fitnesses in reality could depend on the behaviour of individuals themselves, and on the behaviour of those they interact with. In other words, if we were to treat Hamilton's rule being valid as the hypothesis, then the hypothesis covers everything, and the null hypothesis is void, as no true way in which fitness would be determined by the behaviour of self and others would go with Hamilton's rule not holding (see also *Nowak et al., 2017*). This would also inevitably hamper a good statistical analysis, because, given that there is no true model for which Hamilton's rule does not hold, there is no meaningful way to define the distribution of inclusive fitnesses (values of $rb - c$) under the null hypothesis, which one then would want to use to test if $rb - c$ really is larger than $0$.

On the other hand, if we think of Hamilton's rule with costs and benefits defined according to the counterfactual method, then estimating costs and benefits in a linear model without considering the possibility of non-linearities would not allow us to uncover violations, because those, as we have seen, require the fitnesses to be non-linear. One solid way to decide that Hamilton's rule is not violated would be to allow for a non-linear model, and reject all non-linearities, in favour of a linear model. That, however, is not what these papers do.

Our observations therefore first of all imply that meaningful empirical tests would have to consider the version of Hamilton's rule that uses the counterfactual method for computing costs and benefits, and that they should use statistical models that allow for non-linearities. Our observations moreover indicate where to look for violations, and where not to look, as there is a variety of settings in which violations are not expected. First of all, one should look at systems where how much being a co-operator instead of a defector contributes to the fitness of others, and takes away from the fitness of oneself, depends on what the others do. Moreover, if we consider a system that is in equilibrium, we would not observe violations if the three 'synergy-conditions' are satisfied, while systems that (also) have anti-synergies do allow for in-equilibrium violations (see *Figure 4*). The presence of polymorphisms, where co-operators and defectors coexist, can be a good indication that such anti-synergies are present.

## Distinct types

We would also not expect violations in cases where the level of help is a continuous trait, the fitness function is smooth, and where the levels of cooperation across individuals are relatively homogeneous, making fitness effects approximately linear. In models with continuous levels of help, populations can then settle at an equilibrium value of cooperation, where Hamilton's rule is satisfied, provided that there are no branching events (see Section 6 in *van Veelen et al., 2017*). What is needed for violations, therefore, is that there are distinct types; co-operators and defectors. Some behaviours or properties are binary by nature; one either jumps into a river in an effort to save someone, or not. Also eating a fellow brood member is an all or nothing trait (*Pfennig et al., 1999*). When the level of cooperation is a continuous trait, one can however also get distinct types to evolve after a branching event (*Doebeli et al., 2004*). Notice that also at a singular point where the system branches, Hamilton's rule, which predicts no change, does not agree with what happens (*Doebeli and Hauert, 2006*).

## Spurious violations

When $b$ and $c$ are estimated linearly, we should not expect violations of Hamilton's rule. This makes it all the more surprising that 4 out of the 12 explicit tests of Hamilton's rule, surveyed in (*Bourke, 2014*), all of which estimate costs and benefits with a linear model, do find apparent violations. A natural question to ask is where those violations come from. One prime candidate has to do with the fact that the result that Hamilton's rule always holds, if costs and benefits are defined using the regression method, depends on using the cooperativenesses of the individuals involved, not just for computing costs and benefits, but also in the formula for relatedness. The relatedness between siblings, for instance, in this version of Hamilton's rule, would be $cov(x_{se}, x_{si})/var(x_{se})$, where $x_{se}$ refers to the cooperativeness of self and $x_{si}$ to the cooperativeness of the sibling. Here it is good to notice that if these variables refer to data, this is best described, not as relatedness itself, but as an estimator of relatedness. If a different estimator of relatedness is used, for instance based on gene

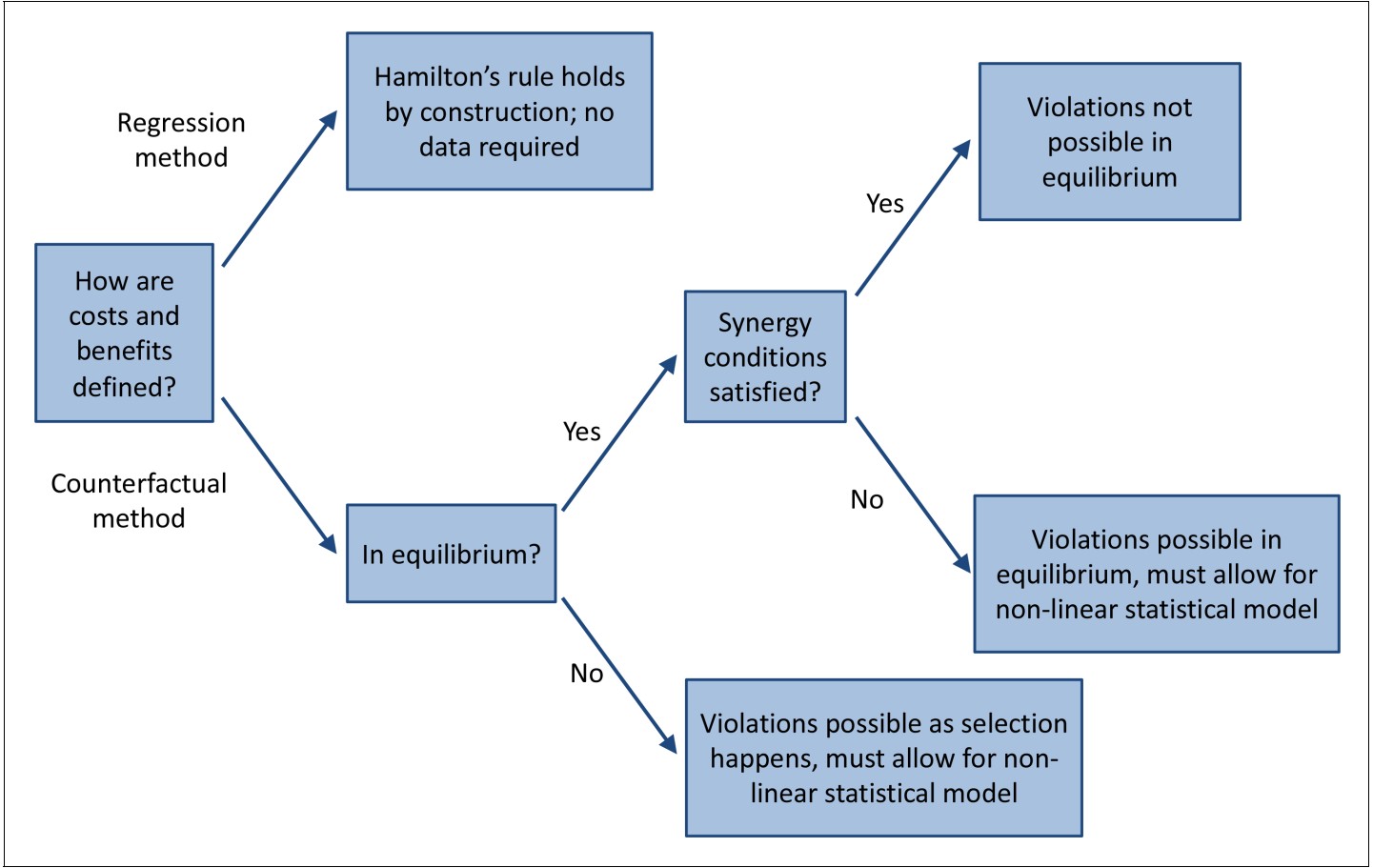

**Figure 4.** A road map for empirical tests of Hamilton's rule. The three synergy conditions are that $\pi_C(i) \geq \pi_C(1) + (i-1)[\pi_D(1) - \pi_D(0)]$ for all $i = 1, ..., n$, that $\pi_D(i) \geq \pi_D(n-1) - (n-i-1)[\pi_C(n) - \pi_C(n-1)]$ for all $i = 0, ..., n-1$, and that $\overline{\pi}_C(p) - \overline{\pi}_D(p)$ increases with $p$.

DOI: https://doi.org/10.7554/eLife.41901.005

sequencing, then that would include genes for traits other than cooperative behaviour. The difference between these two estimators can make for a violation. The probability of this happening decreases as sample size increases. For all sample sizes, large and small, Hamilton's rule, using $cov(x_{se}, x_{si})/var(x_{se})$ for relatedness, will hold. With small samples, costs, benefits, and relatedness may all vary quite a bit – although only in concert, because Hamilton's rule will still have to hold – and with $cov(x_{se}, x_{si})/var(x_{se})$ possibly being far away from the true relatedness, the discrepancy between it and another estimator for relatedness might be large enough to create an apparent violation of Hamilton's rule. Other reasons for finding violations when estimating costs and benefits linearly are also possible (see Section 8 of *van Veelen et al., 2017*), and all of them are spurious.

## Smith et al., 2010

In order to be able to find violations of Hamilton's rule, with benefits and costs defined according to the counterfactual method, one must allow for a non-linear statistical model. The study that comes closest to that ideal is *Smith et al., 2010* (see also *Chuang et al., 2009*; *Chuang et al., 2010*). This experiment does a non-linear estimation of sporulation efficiency of *Myxococcus xanthus*, resulting in fitness functions that would imply that, already in a well-mixed population, co-operators can invade defectors, and defectors can invade co-operators. Combined with a range of population structures, this would make a population settle at mixed overall frequencies of co-operators. This study however also has a few conceptual imperfections. Most of the non-linearity it picks up, results from considering Wrightian fitness's (numbers of offspring) rather than Malthusian fitness's (growth rates); see *Wu et al. (2013)*. For small fitness effects that would not matter too much, but the

aggregate fitness effects are not small here. Another drawback is that the experiment does not include an independent observation of a population structure. For the experiment, a combination of group compositions is chosen. The growth rates at these different group compositions are used to estimate the fitness function – which it is perfect for. The group compositions themselves, as chosen by the experimenter, are also the sole input for calculations characterizing population structure, which makes it not a proper empirical observation of a population structure. In spite of these drawbacks, this study is the closest to a proper empirical test of Hamilton's rule.

### Polymorphisms

While systems of microorganisms with mixed equilibria seem good candidates, one will have to be careful with identifying the reasons for polymorphisms. In systems where one type can invade the other, and vice versa, already in a well-mixed population (*Smith et al., 2010*; *MacLean and Gudelj, 2006*; *MacLean et al., 2010*; *Gore et al., 2009*), the reason for stable coexistence may have nothing to do with population structure. If fitness is moreover maximized at intermediate mixtures, it also hard to unambiguously qualify one strain as cooperative and the other as defecting (*MacLean et al., 2010*). Systems like this might nonetheless point to possible candidates, if there is in fact structure in the population. With microorganisms, one would then naturally want to switch from counting discrete numbers of co-operators to a continuous variable for group compositions, measuring the within group frequency of co-operators. Appendix 1 provides a continuous version that allows for all the equivalent computations needed to establish whether or not Hamilton's rule is violated empirically.

### Wang and Lu, 2018

A study that does look at polymorphisms is *Wang and Lu (2018)*. In this study, players have one of two different roles; there are breeding pairs, which are the potential recipients of help, and potential helpers. This implies that there is no room for the strategic interaction that could create the non-linearities needed for possible violations; costs and benefits of helping at the nest cannot depend on the recipient type, because the recipient is not facing the same choice. Also in this study, therefore, there is no scope for violations of Hamilton's rule.

## Acknowledgements

I would like to thank Peter Spreij, Ben Allen, Carl Veller, and the reviewers.

## Additional information

### Funding
The authors declare that there was no funding for this work.

### Author contributions

Matthijs van Veelen, Conceptualization, Formal analysis, Writing—original draft, Writing—review and editing

### Author ORCIDs

Matthijs van Veelen [iD] http://orcid.org/0000-0002-8290-9212

### Decision letter and Author response
Decision letter https://doi.org/10.7554/eLife.41901.016
Author response https://doi.org/10.7554/eLife.41901.017

## Additional files

### Supplementary files
• Transparent reporting form
DOI: https://doi.org/10.7554/eLife.41901.006

### Data availability

This is a theoretical paper and all relevant information is provided in the manuscript and supporting files.

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

## Appendix 1

DOI:

### I. Introduction

If one would want to test the validity of Hamilton's rule empirically, it would be essential to know if Hamilton's rule can be violated at all, and if it can, it would be essential to know what violations would look like. Whether or not violations are theoretically possible depends on how costs and benefits are defined. Violations are not possible if the costs and benefits in Hamilton's rule are defined according to the "regression method". In that case there can only be spurious reasons for finding social behaviour that is selected for, while the data would seem to suggest that $br<c$, or vice versa. The version of Hamilton's rule that uses the regression method is also referred to as the general version of Hamilton's rule, or HRG; see *Birch (2014)*.

If costs and benefits are defined according to the "counterfactual method", violations are possible. This is the method we will use here, for a variety of reasons. One is that is seems true to the original idea to define for instance the costs of a social act as the difference between an individual's fitness if it performs the social act, and its fitness if does not. A reason not to use the regression method is also that this method is incomplete, and as such not well-defined; it ignores the issue of model specification, and therefore there are cases where different specifications would result in different Hamilton's rules, with different values for costs and benefits; see the main text and Section 4 of *van Veelen et al. (2017)*. While the version of Hamilton's rule that uses the regression method is also referred to as HRG by *Birch, 2014*, the version of Hamilton's rule that uses the counterfactual method is *not* what *Birch, 2014* calls the special version of Hamilton's rule, of HRS. See Section 9 of *van Veelen et al., 2017* for details.

Although the counterfactual method allows for violations, it off course also allows for non-violations. If the interaction is characterized by 'equal gains from switching' then violations should not occur. Equal gains from switching means that the fitness effects (the costs and benefits) of the social behaviour are independent of who else contributes, and also independent of whether or not the recipient performs the behaviour. With equal gains from switching, the costs and benefits therefore are also independent of the current share of co-operators in the population. Violations are only possible if the interaction does not have equal gains from switching, and in that case costs and benefits will vary with the current frequency of co-operators.

The fact that cost and benefits are frequency-dependent in those cases can make it harder to derive claims that hold for all frequencies. We will therefore begin by focusing on two specific frequencies: $p = 0$ and $p = 1$. At, or close to those frequencies, we will compare the direction of selection – which here will indicate whether or not one strategy can invade the other, and vice versa – with the value of $rb - c$ at those frequencies. For that, it will be useful to define a "population structure profile" at $p = 0$, and one at $p = 1$. This will be useful for rewriting both inclusive fitness and the direction of selection, and for comparing these to each other. Later on, when looking at cases with equilibrium mixtures of co-operators and defectors, we will also consider frequencies $p$ in between 0 and 1, and revisit this more generally.

### II. Relatedness at the edges

#### A. Close to $p = 0$

Performing a cooperative act is assumed to be costly. Not all cooperative acts are costly, because there are also mutualistic forms of cooperation, but in this context, we will mostly focus on costly cooperation, hence the assumption. The benefits of being a co-operator – benefits that may or may not outweigh the costs – we assume comes from the structure in the population. This structure might make co-operators also be on the receiving end more often,

as they might be matched with co-operators more often than defectors are. This difference is captured by relatedness. Relatedness is defined as the difference in probabilities (between a co-operator and a defector) of being matched with a co-operator: $P(C|C) - P(C|D)$. Both of these conditional probabilities will typically depend on the current overall frequency of co-operators, denoted by $p$, but for a variety of population structures, the difference between them will not depend on $p$.

In order to compute that difference, we think of a simple chance experiment. First draw an individual from the population, with every individual being equally likely to be drawn. Then randomly select a partner. In our case, we will use a simple setup, where individuals live in groups of equal size. In this case, the second draw simply returns to the group that the first draw happened to come from, and draws a random other group member, with all (other) group members having equal probability of being drawn.

In this setup with groups of equal size, a population structure is defined by functions $f_i(p)$, $i = 0, ..., n$, that specify the frequency of groups with $i$ co-operators and $n - i$ defectors in it, at an overall frequency $p$ of co-operators. Using this notation, and Bayes' rule, the difference in conditional probabilities can be written as

$$r(p) = P(C|C) - P(C|D) = \frac{\sum_{i=0}^n \frac{i}{n}\frac{i-1}{n-1}f_i(p)}{\sum_{i=0}^n \frac{i}{n}f_i(p)} - \frac{\sum_{i=0}^n \frac{n-i}{n}\frac{i}{n-1}f_i(p)}{\sum_{i=0}^n \frac{n-i}{n}f_i(p)}$$

Close to $p = 0$, the second term on the right hand side goes to 0, because if there are almost no co-operators around, the chance that a random defector runs into a co-operator must go to 0. In other words, the enumerator goes to 0, because

$$\lim_{p\downarrow 0} f_i(p) = \begin{cases} 0 & \text{if } i>0 \\ 1 & \text{if } i = 0 \end{cases}$$

and because $f_0(p)$ is multiplied by $\frac{0}{n-1}$. These limits for $p \downarrow 0$ follow from the fact that $\sum_{i=0}^n \frac{i}{n}f_i(p) = p$; this implies that $\frac{i}{n}f_i(p)<p$ for each $i>0$, and hence $f_i(p)<\frac{n}{i}p$. Therefore $\lim_{p\downarrow 0} f_i(p) = 0$ for $i>0$, and since $\sum_{i=0}^n f_i(p) = 1$, moreover $\lim_{p\downarrow 0} f_0(p) = 1$.

The denominator goes to 1 because

$$\lim_{p\downarrow 0} \sum_{i=0}^n \frac{n-i}{n} f_i(p) = \lim_{p\downarrow 0} (1-p) = 1$$

Therefore we can write

$$\lim_{p\downarrow 0} r(p) = \lim_{p\downarrow 0} P(C|C) = \lim_{p\downarrow 0} \frac{\sum_{i=0}^n \frac{i}{n}\frac{i-1}{n-1}f_i(p)}{\sum_{i=0}^n \frac{i}{n}f_i(p)} = \lim_{p\downarrow 0} \frac{\sum_{i=0}^n \frac{i}{n}\frac{i-1}{n-1}f_i(p)}{p}$$

While relatedness is a one-dimensional measure that characterizes a population state, or a population structure to some extent, a richer measure will be useful too. This measure captures how likely co-operators are to find themselves in groups of different types at low frequencies $p$.

$$\underline{u}_i = \lim_{p\downarrow 0} \frac{\frac{i}{n}f_i(p)}{\sum_{j=0}^n \frac{j}{n}f_j(p)} = \lim_{p\downarrow 0} \frac{\frac{i}{n}f_i(p)}{p}, i = 0, ..., n$$

With this definition, $\underline{u}_i$ is the share of mutant co-operators that find themselves in a group with $i$ co-operators. Relatedness then becomes the inner product of a vector $\underline{\mathbf{u}}$ with $\underline{u}_i$ as its $i$th element, and a vector with $\frac{i-1}{n-1}$ as its $i$th element:

$$\lim_{p\downarrow 0} r(p) = \sum_{i=0}^n \underline{u}_i \frac{i-1}{n-1} = \sum_{i=2}^n \underline{u}_i \frac{i-1}{n-1}$$

The vector $\underline{\mathbf{u}}$ could be labelled the 'population structure profile' in the limit of $p \downarrow 0$, and it is clear that there is typically a variety of profiles $\underline{\mathbf{u}}$ that produces one and the same relatedness.

## B. Close to $p = 1$

Close to $p = 1$, it is easier to use the equivalent definition of relatedness, reversing the roles of co-operators and defectors:

$$r(p) = P(D|D) - P(D|C) = \frac{\sum_{i=0}^{n} \frac{n-i}{n} \frac{n-i-1}{n-1} f_i(p)}{\sum_{i=0}^{n} \frac{n-i}{n} f_i(p)} - \frac{\sum_{i=0}^{n} \frac{i}{n} \frac{n-i}{n-1} f_i(p)}{\sum_{i=0}^{n} \frac{i}{n} f_i(p)}$$

The second term on the right hand side goes to 0 again; the enumerator goes to 0, because

$$\lim_{p \uparrow 1} f_i(p) = \begin{cases} 0 & \text{if } i < n \\ 1 & \text{if } i = n \end{cases}$$

and because $f_n(p)$ is multiplied by $\frac{0}{n-1}$, while the denominator goes to 1 because

$$\lim_{p \uparrow 1} \sum_{i=0}^{n} \frac{i}{n} f_i(p) = \lim_{p \uparrow 1} p = 1$$

Therefore we can write

$$\lim_{p \uparrow 1} r(p) = \lim_{p \uparrow 1} P(D|D) = \lim_{p \uparrow 1} \frac{\sum_{i=0}^{n} \frac{n-i}{n} \frac{n-i-1}{n-1} f_i(p)}{\sum_{i=0}^{n} \frac{n-i}{n} f_i(p)} = \lim_{p \uparrow 1} \frac{\sum_{i=0}^{n} \frac{n-i}{n} \frac{n-i-1}{n-1} f_i(p)}{1 - p}$$

We can furthermore define a measure that captures how likely defectors are to find themselves in groups of different types at low frequencies $1 - p$ of defectors (which means high frequencies $p$ of co-operators).

$$\bar{u}_i = \lim_{p \uparrow 1} \frac{\frac{n-i}{n} f_i(p)}{\sum_{j=0}^{n} \frac{n-j}{n} f_j(p)} = \lim_{p \uparrow 1} \frac{\frac{n-i}{n} f_i(p)}{1 - p}, \quad i = 0, ..., n$$

Here $\bar{u}_i$ is the share of mutant defectors that finds itself in a group with $i$ co-operators. In many models it would be reasonable to assume that mutant defectors face similar numbers of fellow defectors in their group as mutant co-operators face fellow co-operators. If they do, the population structure profiles at $p = 0$ and $p = 1$ would be each other's mirror images: $\underline{u}_i = \bar{u}_{n-i}$. Relatedness at $p = 1$ again becomes the inner product of a vector $\bar{u}$ with $\bar{u}_i$ as its $i$th element, and a vector with $\frac{n-i-1}{n-1}$ as its $i$th element.

$$\lim_{p \uparrow 1} r(p) = \sum_{i=0}^{n} \bar{u}_i \frac{n-i-1}{n-1} = \sum_{i=0}^{n-2} \bar{u}_i \frac{n-i-1}{n-1}$$

If indeed $\underline{u}_i = \bar{u}_{n-i}$, then $\lim_{p \uparrow 1} r(p) = \lim_{p \downarrow 0} r(p)$.

   These population structure profiles, $\underline{u}$ at $p = 0$ and $\bar{u}$ at $p = 1$, will be useful for rewriting when cooperation can invade defection ($p = 0$), and when defection can invade cooperation ($p = 1$), as well as when inclusive fitness is positive or negative at these two frequencies.

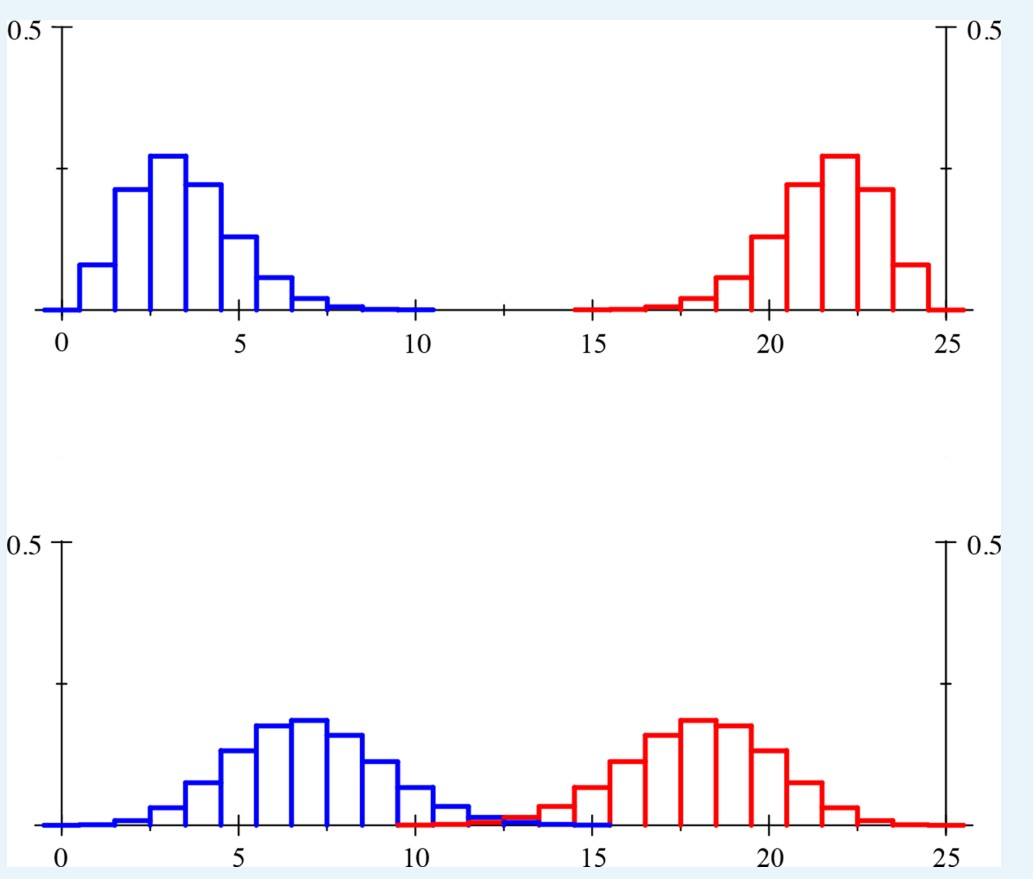

**Appendix 1—figure 1.** Population structure profiles $\underline{u}$ (blue) for co-operator invaders at $p \downarrow 0$, and $\overline{u}$ (red) for defector invaders at $p \uparrow 1$. Defectors at $p \downarrow 0$ only find themselves in groups with $0$ defectors, co-operators at $p \uparrow 1$ only find themselves in groups with $n$ co-operators (not depicted in the figures). In the lower panel mutants assort more than in the upper one, and random matching would result in a population structure profile $\overline{u}$ with single spike at $1$, and a single spike at $n-1$ for $\overline{u}$. Section IX describes the population structures that result in these population structure profiles. For this population structure $\underline{u}_i = \overline{u}_{n-i}$ is satisfied, which means that the red bars are the mirror image of the blue ones.

DOI: https://doi.org/10.7554/eLife.41901.008

## III. When cooperation/defection can/cannot invade

### A. Close to $p = 0$

Mutant $C$'s can invade a resident $D$ if they have higher fitness close to $p = 0$:

$$\lim_{p \downarrow 0} \overline{\pi}_C > \lim_{p \downarrow 0} \overline{\pi}_D$$

The game between co-operators and defectors is defined by $\pi_C(i)$ for co-operators and $\pi_D(i)$ for defectors that determine the payoff they get as a function of the number of co-operators in the group they are in (for co-operators: including themselves). We assume a dynamic that linearly translates payoffs into fitness, like the replicator dynamics. This allows us to use payoffs and fitness interchangeably; see *van Veelen, 2011*.

In the limit of $p \downarrow 0$, all defectors find themselves in groups with other defectors only, whatever the population structure is, which makes $\lim_{p \downarrow 0} \overline{\pi}_D = \pi_D(0)$. The condition for a $C$ invader to do better than a $D$ resident then becomes:

$$\lim_{p \downarrow 0} \frac{\sum_{i=1}^{n} \frac{i}{n} f_i(p) \pi_C(i)}{\sum_{i=1}^{n} \frac{i}{n} f_i(p)} > \pi_D(0)$$

With the shorthand notation introduced in Section II, this can be written as:

$$\sum_{i=1}^{n} \underline{u}_i \pi_C(i) > \pi_D(0)$$

where $\underline{u}_i$ is the share of mutant co-operators that find themselves in groups that contain $i$ co-operators.

## B. Close to $p = 1$

Mutant $D$'s cannot invade a resident $C$ if the resident has higher fitness at $p = 1$:

$$\lim_{p \uparrow 1} \overline{\pi}_C > \lim_{p \uparrow 1} \overline{\pi}_D$$

In the limit of $p \uparrow 1$, all co-operators find themselves in groups with only co-operators, which makes $\lim_{p \uparrow 1} \overline{\pi}_C = \pi_C(n)$. The condition for $C$ to drive out $D$ invaders then becomes:

$$\pi_C(n) > \lim_{p \uparrow 1} \frac{\sum_{i=0}^{n-1} \frac{n-i}{n} f_i(p) \pi_D(i)}{\sum_{i=0}^{n-1} \frac{n-i}{n} f_i(p)}$$

With the shorthand notation introduced above, this can be written as:

$$\pi_C(n) > \sum_{i=0}^{n-1} \overline{u}_i \pi_D(i)$$

where $\overline{u}_i$ is the share of mutant defectors that find themselves in groups that contain $i$ co-operators.

# IV. When inclusive fitness is positive/negative

## A. Close to $p = 0$

At full defection ($p = 0$), inclusive fitness is positive if:

$$\lim_{p \downarrow 0} r(p) b(p) > c(p)$$

The costs of cooperation at $p = 0$ – according to the counterfactual method – are $\pi_D(0) - \pi_C(1) = -(\pi_C(1) - \pi_D(0))$. This is minus the difference between the payoff a co-operator gets in a group where everyone else is a defector, and the payoff gets if it were a defector instead. The aggregate benefits are $(n-1)[\pi_D(1) - \pi_D(0)]$, which is $n-1$ times how much every defector in that group benefits from the agent being a co-operator instead of a defector. Inclusive fitness being positive at $p = 0$ therefore can be rewritten as

$$\lim_{p \downarrow 0} \frac{\sum_{i=1}^{n} \frac{i}{n} \frac{i-1}{n-1} f_i(p)}{\sum_{i=1}^{n} \frac{i}{n} f_i(p)} (n-1)[\pi_D(1) - \pi_D(0)] > \pi_D(0) - \pi_C(1)$$

With a little rewriting, this inequality becomes

$$\pi_C(1) + \lim_{p \downarrow 0} \frac{\sum_{i=1}^{n} i(i-1) f_i(p)}{\sum_{i=1}^{n} i f_i(p)} [\pi_D(1) - \pi_D(0)] > \pi_D(0)$$

With $\frac{\sum_{i=1}^{n} i f_i(p)}{\sum_{i=1}^{n} i f_i(p)} = 1$, this can be rewritten as:

$$\lim_{p\downarrow 0} \frac{\sum_{i=1}^{n} i f_i(p)\{\pi_C(1) + (i-1)[\pi_D(1) - \pi_D(0)]\}}{\sum_{i=1}^{n} i f_i(p)} > \pi_D(0)$$

In shorthand notation, inclusive fitness at $p = 0$ is positive if:

$$\sum_{i=1}^{n} \underline{u}_i \{\pi_C(1) + (i-1)[\pi_D(1) - \pi_D(0)]\} > \pi_D(0)$$

## B. Close to $p = 1$

At full cooperation ($p = 1$), inclusive fitness is positive if:

$$\lim_{p\uparrow 1} r(p)b(p) > c(p)$$

The costs of cooperation at $p = 1$ – according to the counterfactual method – are $\pi_D(n-1) - \pi_C(n) = -(\pi_C(n) - \pi_D(n-1))$. This is minus the difference between the payoff a co-operator gets in a group where everyone else is a co-operator too, and the payoff it gets if it were a defector instead. The aggregate benefits are $(n-1)[\pi_C(n) - \pi_C(n-1)]$, which is $n-1$ times how much every other co-operator in that group benefits from the agent being a co-operator instead of a defector. Inclusive fitness being positive at $p = 0$ therefore can be rewritten as

$$\lim_{p\uparrow 1} \frac{\sum_{i=0}^{n-1} \frac{n-i}{n}\frac{n-i-1}{n-1} f_i(p)}{\sum_{i=0}^{n-1} \frac{n-i}{n} f_i(p)} (n-1)[\pi_C(n) - \pi_C(n-1)] > \pi_D(n-1) - \pi_C(n)$$

With a little rewriting, this inequality becomes

$$\pi_C(n) > \pi_D(n-1) - \lim_{p\uparrow 1} \frac{\sum_{i=0}^{n-1} (n-i)(n-i-1) f_i(p)}{\sum_{i=0}^{n-1} (n-i) f_i(p)} [\pi_C(n) - \pi_C(n-1)]$$

With $\frac{\sum_{i=0}^{n-1} (n-i) f_i(p)}{\sum_{i=0}^{n-1} (n-i) f_i(p)} = 1$, this can be rewritten as:

$$\pi_C(n) > \lim_{p\uparrow 1} \frac{\sum_{i=0}^{n-1} (n-i) f_i(p)\{\pi_D(n-1) - (n-i-1)[\pi_C(n) - \pi_C(n-1)]\}}{\sum_{i=0}^{n-1} (n-i) f_i(p)}$$

In shorthand notation, inclusive fitness at $p = 1$ is positive if:

$$\pi_C(n) > \sum_{i=0}^{n-1} \overline{u}_i \{\pi_D(n-1) - (n-i-1)[\pi_C(n) - \pi_C(n-1)]\}$$

## V. (Not) observing violations in equilibrium

Now that we have the criteria, both at $p = 0$ and at $p = 1$, for co-operators to be selected for, as well as for inclusive fitness to be positive, we can compare them to each other, and find out what it takes to observe a violation.

At $p = 0$, co-operators can invade if $\sum_{i=1}^{n} \underline{u}_i \pi_C(i) > \pi_D(0)$. If we observe a population at $p = 0$, and if we assume, or have reason to believe, that this population is in equilibrium, then that means that co-operators can apparently not invade at $p = 0$, and the opposite is true:

$$\sum_{i=1}^{n} \underline{u}_i \pi_C(i) < \pi_D(0) \tag{1}$$

Inclusive fitness at $p = 0$ is negative if

$$\sum_{i=1}^{n} \underline{u}_i \{\pi_C(1) + (i-1)[\pi_D(1) - \pi_D(0)]\} < \pi_D(0) \tag{2}$$

Now there is a variety of ways in which the left hand sides of these two equations can be different from each other, while they are still both smaller than the right hand side that they share. In that case inclusive fitness is not the right criterion, but we would still not observe a violation in equilibrium. This will most certainly be the case if each and every term in the second equation that replaces a $\pi_C(i)$–term from the first one is smaller, making the left hand side of the second equation even smaller than the left hand side of the first. In other words, a sufficient condition for the left hand side of the second equation to be lower than, or at most equal to, the left hand side of the first, is that

$$\pi_C(1) + (i-1)[\pi_D(1) - \pi_D(0)] \le \pi_C(i) \tag{3}$$

for all $i \in 1, ..., n$. This would then imply that

$$\sum_{i=1}^{n} \underline{u}_i \{\pi_C(1) + (i-1)[\pi_D(1) - \pi_D(0)]\} \le \sum_{i=1}^{n} \underline{u}_i \pi_C(i) < \pi_D(0)$$

and therefore that inclusive fitness is certainly negative whenever cooperation is selected against. A $\pi_C$ and $\pi_D$ for which this is true thereby precludes observing a violation of Hamilton's rule in equilibrium at $p = 0$.

Observing violations in equilibrium at $p = 0$ would require that at least for one $i$,

$$\pi_C(1) + (i-1)[\pi_D(1) - \pi_D(0)] > \pi_C(i)$$

and that these differences are sufficiently large to make the left hand side of **Equation 2** larger than the right hand side, leading to a positive inclusive fitness, while the behaviour is selected against.

This is mirrored at $p = 1$, where defectors cannot invade if

$$\pi_C(n) > \sum_{i=0}^{n-1} \overline{u}_i \pi_D(i) \tag{4}$$

Inclusive fitness at $p = 1$ is positive if

$$\pi_C(n) > \sum_{i=0}^{n-1} \overline{u}_i \{\pi_D(n-1) - (n-i-1)[\pi_C(n) - \pi_C(n-1)]\} \tag{5}$$

Again there is a variety of ways in which the right hand sides of these two equations can be different from each other, while they are both smaller than the left hand side that they share. In that case inclusive fitness is not the right criterion, but we would not observe a violation in equilibrium. That will certainly be the case if the game is such that

$$\pi_D(n-1) - (n-i-1)[\pi_C(n) - \pi_C(n-1)] \le \pi_D(i) \tag{6}$$

for all $i \in 0, ..., n-1$. This would be a sufficient condition for the right hand side of the second equation to be lower than, or at most equal to, the right hand side of the first, so that

$$\pi_C(n) > \sum_{i=0}^{n-1} \overline{u}_i \pi_D(i) \ge \sum_{i=0}^{n-1} \overline{u}_i \{\pi_D(n-1) - (n-i-1)[\pi_C(n) - \pi_C(n-1)]\}$$

For games that satisfy this condition, observing violations in equilibrium at $p = 1$ therefore is impossible, because inclusive fitness is certainly positive if cooperation is selected for.

Observing violations in equilibrium at $p = 1$ would require that at least for one $i$,

$$\pi_D(n-1) - (n-i-1)[\pi_C(n) - \pi_C(n-1)] > \pi_D(i)$$

and that these differences are sufficiently large to make the right hand side of *Equation 5* larger than the left hand side, leading to a negative inclusive fitness, while the behaviour is selected for.

## A. Not observing violations in equilibrium, also in the interior

While inequalities (*Equation 3*) and (*Equation 6*) preclude violations of Hamilton's rule at $p = 0$ and $p = 1$, respectively, they do not necessarily also preclude violations in the interior. In order to see why in general they do not, we can first look at a simple special case where they do, and that is if $n = 2$, and if, moreover, relatedness is constant.

For $n = 2$, inequalities (*Equation 3*) and (*Equation 6*) are the same; they both require $\pi_D(1) - \pi_D(0)$ to be less or equal to $\pi_C(2) - \pi_C(1)$. If these two differences would be equal, then that would be the classical 'equal gains from switching', so one could perhaps call it 'increasing gains from switching' if the latter is larger.

A population structure with constant relatedness and $n = 2$ can only be

$$
\begin{aligned}
f_0(p) &= (1-r)(1-p)^2 + r(1-p) \\
f_1(p) &= (1-r)2p(1-p) \\
f_0(p) &= (1-r)p^2 + rp
\end{aligned}
$$

We can compute and rewrite the average payoffs of co-operators and defectors for $n = 2$ as follows:

$$
\begin{aligned}
\overline{\pi}_C(p) &= \pi_C(2)[(1-r)p + r] + \pi_C(1)[(1-r)(1-p)] \\
&= r\pi_C(2) + (1-r)\pi_C(1) + (1-r)p[\pi_C(2) - \pi_C(1)]
\end{aligned}
$$

and

$$
\begin{aligned}
\overline{\pi}_D(p) &= \pi_D(0)[(1-r)(1-p) + r] + \pi_D(1)[(1-r)p] \\
&= \pi_D(0) + (1-r)p[\pi_D(1) - \pi_D(0)]
\end{aligned}
$$

Since the terms between square brackets in the second line of both are the 'gains from switching', increasing gains from switching here implies that the difference in average fitness between co-operators and defectors $\overline{\pi}_C(p) - \overline{\pi}_D(p)$ is increasing in the frequency of co-operators $p$. Therefore, at $n = 2$, inequality (*Equation 3*) and/or (*Equation 6*), combined with constant relatedness, implies that if co-operators are selected for at $p = 0$, they will be selected all the way to $p = 1$, and if they are selected against at $p = 1$, they will be selected against all the way to $p = 0$. This implies that there are no stable interior equilibria, and therefore are there not only no in-equilibrium violations possible at $p = 0$ and $p = 1$, but also not in between. This is illustrated in *Figure 3a* in the main text.

If we go to group sizes $n$ larger than 2, there are many degrees of freedom that make things much more complicated. First of all, conditions (*Equation 3*) and (*Equation 6*) do not imply increased gains from switching in some more general, across the board sense. Inequality (*Equation 3*) only puts limitations on how $\pi_C(i) - \pi_C(1)$ relates to $\pi_D(1) - \pi_C(0)$, but not, for instance, on how $\pi_C(i) - \pi_C(i - 1)$ relates to $\pi_C(i - 1) - \pi_C(i - 2)$ or to $\pi_D(i - 1) - \pi_D(i - 2)$. One can therefore construct payoff functions that satisfy both inequalities, and yet have gains from switching that sometimes increase and sometimes decrease, depending on how many co-operators there currently are.

Second, one can also tinker with the population structure to create internal stable equilibria. While the grey lines in *Figure 3a* in the main text represent population structures with fixed $r$, it is clear that if $r$ is allowed to vary with $p$, one can create grey lines (population structures) that have multiple intersections with the green line that separates the region where co-operators are selected for and selected against, thereby creating stable and unstable equilibria. This can happen, in spite of the fact that, for $n = 2$, inequalities (*Equation 3*) and (*Equation 6*) are as global as it gets. Moreover, with $n > 2$ there are enough degrees of freedom in the notion of a population structure to expect that, when combined with almost any fitness function, one might be able to cook up populations structures that result in stable interior equilibria, even with constant $r$. In any case it is clear that conditions on the fitness

function alone are not enough to preclude violations of Hamilton's rule in stable interior equilibria.

For many examples, however, it is more straightforward to directly check the key property that would be relevant to preclude stable interior equilibria, and that is that population structure and fitness function combine in a way that makes $\overline{\pi}_C(p) - \overline{\pi}_D(p)$ non-decreasing in $p$ – as we did for the case of $n = 2$ and fixed relatedness. If inequalities (**Equation 3**) and (**Equation 6**) are satisfied, and this 'relative synergies' property also holds, then all in-equilibrium violations of Hamilton's rule are ruled out. Just to make sure: this relative synergies property is symmetric, in the sense that $\overline{\pi}_C(p) - \overline{\pi}_D(p)$ is non-decreasing in $p$ if and only if $\overline{\pi}_D(p) - \overline{\pi}_C(p)$ is non-decreasing in $1 - p$.

## VI. Examples

The two conditions from Section V, one for $p = 0$, one for $p = 1$, allow us to easily make examples of strategic interactions that do not allow for any observations that violate Hamilton's rule, as long as the population is in equilibrium. Take for instance payoff functions of the following form:

$$\begin{aligned} \pi_C(i) &= 1 + g(i) - K \\ \pi_D(i) &= 1 + g(i) \end{aligned}$$

This can be interpreted as a situation where co-operators can produce a public good. How much is produced depends on how many individuals contribute, and that is reflected by $g(i)$. Besides affecting the payoffs through the public good, contributing always lowers the payoff of an agent by a fixed amount $K$, which is why co-operators always get a payoff that is $K$ below what their defecting fellow group members get. (The costs, according to the counterfactual method, are typically of the form $\pi_D(i) - \pi_C(i+1) = K + (g(i) - g(i+1))$. These also take the difference in payoffs due to the public good production into account, of which the agent itself also benefits, and those are not assumed to be constant.)

A sufficient condition on $g(i)$ to prevent violations of Hamilton's rule in equilibrium at $p = 0$ and $p = 1$ is that there should be synergies – or at least no dis-synergies – in the production of the public good:

$$g(i) - g(i-1) \geq g(i-1) - g(i-2)$$

for $i = 1, ..., n$. This implies that

$$\begin{aligned} \pi_C(1) + (i-1)[\pi_D(1) - \pi_D(0)] &= \\ 1 + g(1) - K + (i-1)[g(1) - g(0)] &\leq 1 + g(i) - K \\ &= \pi_C(i) \end{aligned}$$

for $i = 1, ..., n$. In Section V we saw that this prevents violations of Hamilton's rule at $p = 0$. Moreover,

$$\begin{aligned} \pi_D(n-1) - (n-i-1)[\pi_C(n) - \pi_C(n-1)] &= \\ 1 + g(n-1) - (n-i-1)[g(n) - g(n-1)] &\leq 1 + g(i) \\ &= \pi_D(i) \end{aligned}$$

for $i = 0, ..., n-1$, which we saw prevents violations of Hamilton's rule at $p = 1$.

A few specific choices for $g(i)$ can also help illustrate a few possible situations.

## Equal gains

If $g(i) = 2\left(\frac{i}{n}\right)$ and $K = 0.4$, then

$$\pi_C(i) = 0.6 + 2\left(\frac{i}{n}\right)$$
$$\pi_D(i) = 1 + 2\left(\frac{i}{n}\right)$$

Combined with a population structure elaborated on in Section IX, this results in $\overline{\pi}_C - \overline{\pi}_D = 0.16$, both at $p = 0$ and at $p = 1$. Inclusive fitness is also $r(p)b(p) - c(p) = \frac{1}{4}\frac{48}{25} - \frac{32}{100} = 0.16$, also both at $p = 0$ and at $p = 1$. Cooperation therefore will be selected for at both extremes (and everywhere in between) and inclusive fitness agrees with the direction of selection. This is always the case with equal gains from switching. Details about the computations can be found in Section IX.

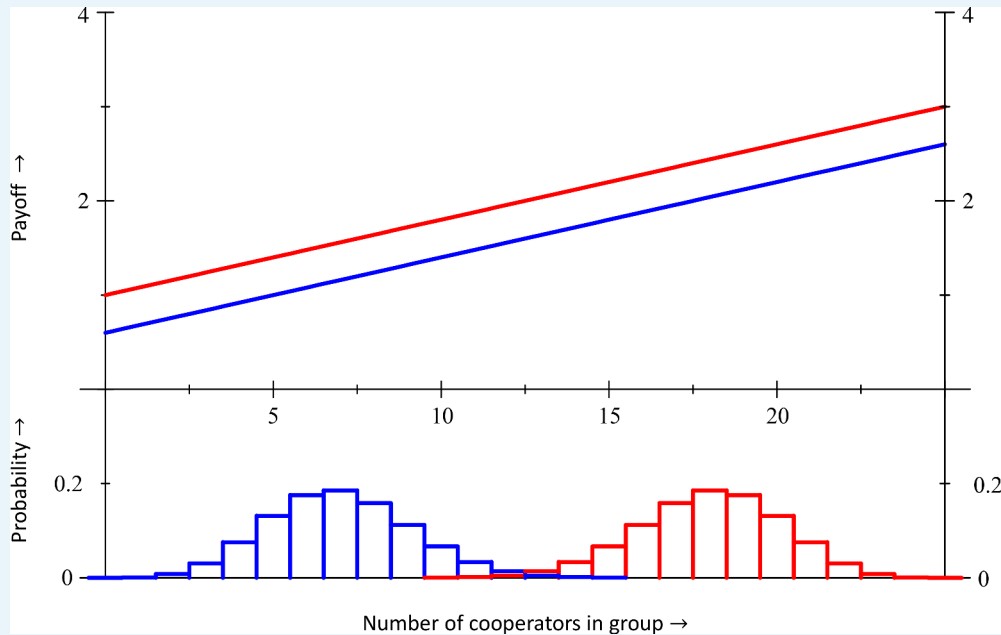

**Appendix 1—figure 2.** The population structure profiles are depicted in the lower half of the figure, the payoffs in the upper half.

DOI: https://doi.org/10.7554/eLife.41901.009

## Synergies

If $g(i) = 2\left(\frac{i}{n}\right)^2$ and $K = 0.5$, then

$$\pi_C(i) = 0.5 + 2\left(\frac{i}{n}\right)^2$$
$$\pi_D(i) = 1 + 2\left(\frac{i}{n}\right)^2$$

Combined with the same population structure elaborated on in Section IX, this results in $\overline{\pi}_C - \overline{\pi}_D = -0.33$ at $p = 0$ and $\overline{\pi}_C - \overline{\pi}_D = 0.45$ at $p = 1$. Cooperation therefore will be selected against at $p = 0$, and selected for at $p = 1$, which makes both $p = 0$ and $p = 1$ stable equilibria. Inclusive fitness is $-0.48$ at $p = 0$ and $0.6$ at $p = 1$. Inclusive fitness therefore is more negative than $\overline{\pi}_C - \overline{\pi}_D$ at $p = 0$, and more positive at $p = 1$, which prevents observing violations at equilibrium. Details of the computations, as well as the computation of costs and benefits for the regression method, can be found in Section IX.

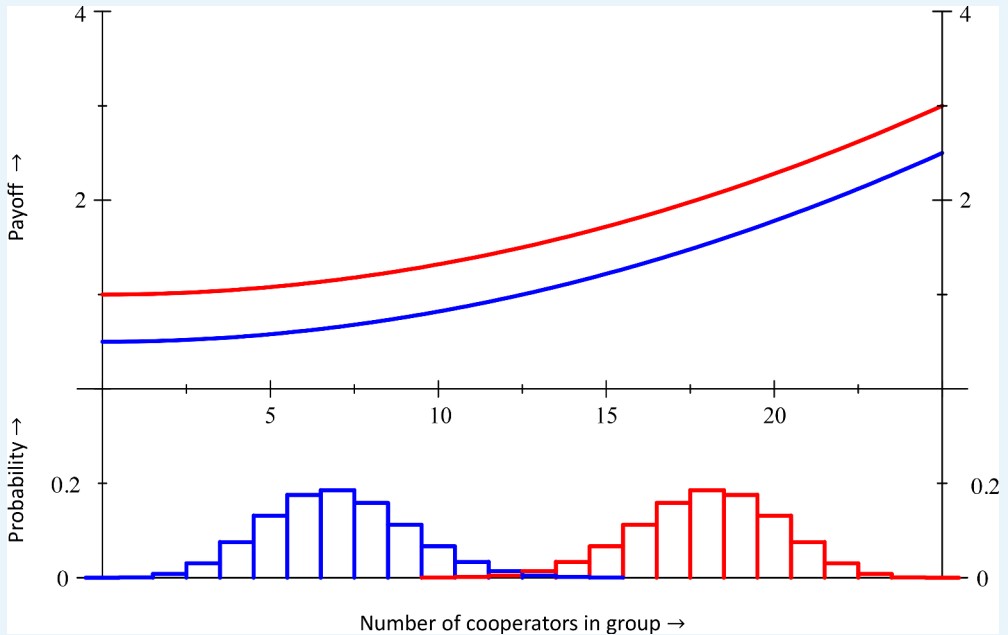

**Appendix 1—figure 3.** The population structure profiles are depicted in the lower half of the figure, the payoffs in the upper half.

DOI: https://doi.org/10.7554/eLife.41901.010

## Anti-synergies

If $g(i) = 2\left(\frac{i}{n}\right)^{\frac{1}{2}}$ and $K = 0.5$, then

$$\pi_C(i) = 0.5 + 2\left(\frac{i}{n}\right)^{\frac{1}{2}}$$
$$\pi_D(i) = 1 + 2\left(\frac{i}{n}\right)^{\frac{1}{2}}$$

Combined with the same population structure elaborated on in Section IX, this results in $\overline{\pi}_C - \overline{\pi}_D = 0.55$ at $p = 0$ and $\overline{\pi}_C - \overline{\pi}_D = -0.19$ at $p = 1$. Cooperation therefore will be selected for at $p = 0$, and selected against at $p = 1$, which makes neither $p = 0$ nor $p = 1$ an equilibrium. Inclusive fitness is 2.3 at $p = 0$ and $-0.22$ at $p = 1$. Inclusive fitness therefore is more positive than $\overline{\pi}_C - \overline{\pi}_D$ at $p = 0$, and more negative than $\overline{\pi}_C - \overline{\pi}_D$ at $p = 1$. This particular combination of a payoff function and a population structure would not lead to observations of violations in equilibrium at $p = 0$ or $p = 1$, but if we simply increase $K$, or choose a population structure with less assortment, we could make $\overline{\pi}_C - \overline{\pi}_D$ drop below 0 at $p = 0$, while inclusive fitness is still positive. At the other end, we could decrease $K$, or increase assortment, to make $\overline{\pi}_C - \overline{\pi}_D$ positive, while inclusive fitness is still below 0, also creating a violation in equilibrium. Moreover, and more importantly, since co-operators invade defectors and defectors invade co-operators, we can expect violations at a mixed equilibrium, which is what Section VII is about. Details of the computations, as well as the computation of costs and benefits for the regression method, can be found in Section IX.

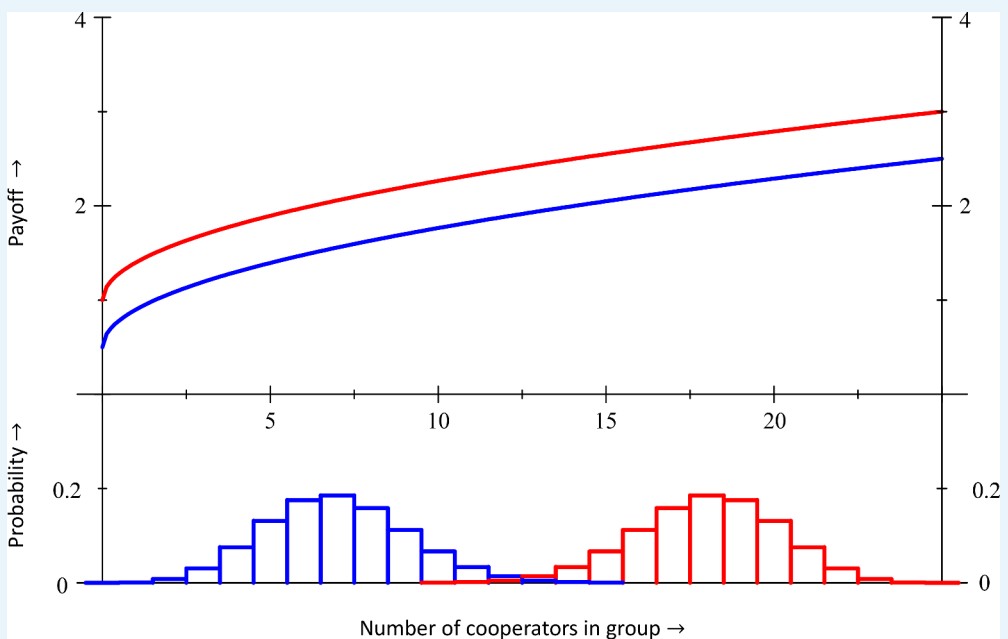

**Appendix 1—figure 4.** The population structure profiles are depicted in the lower half of the figure, the payoffs in the upper half.

DOI: https://doi.org/10.7554/eLife.41901.011

## VII. Mixed equilibria

When co-operators can invade defectors and defectors can invade co-operators, there for sure must be at least one equilibrium mixture of co-operators and defectors. Also other games without equal gains from switching may have stable frequencies $p$ at which co-operators and defectors coexist. At those frequencies, we can compare the direction of selection with inclusive fitness. In Sections III, IV and V we did this comparison at $p = 0$ and $p = 1$ only, now we will need to also do it at the equilibrium frequency, which may be anywhere between 0 and 1.

## A. Population structure profiles and relatedness

Similar to before, we can define the probability that a randomly chosen co-operator finds itself in a group in which there are $i$ co-operators:

$$u_{i,C}(p) = \frac{\frac{i}{n}f_i(p)}{\sum_{j=0}^{n}\frac{i}{n}f_j(p)} = \frac{\frac{i}{n}f_i(p)}{p}$$

and the probability that a randomly chosen defector finds itself in a group in which there are $i$ co-operators:

$$u_{i,D}(p) = \frac{\frac{n-i}{n}f_i(p)}{\sum_{j=0}^{n}\frac{n-j}{n}f_j(p)} = \frac{\frac{n-i}{n}f_i(p)}{1-p}$$

This makes $\underline{u}_i = \lim_{p\downarrow 0} u_{i,C}(p)$ and $\overline{u}_i = \lim_{p\uparrow 1} u_{i,D}(p)$. Relatedness now becomes

$$r(p) = \sum_{i=0}^{n} u_{i,C}(p)\frac{i-1}{n-1} - \sum_{i=0}^{n} u_{i,D}(p)\frac{i}{n-1}$$

## B. Equilibrium

An interior equilibrium is a frequency $p^* \in (0,1)$ such that the average payoff of co-operators and the average payoff of defectors are equal:

$$\overline{\pi}_C(p^*) = \overline{\pi}_D(p^*)$$

This can be written as

$$\sum_{i=0}^{n} u_{i,C}(p^*)\pi_C(i) = \sum_{i=0}^{n} u_{i,D}(p^*)\pi_D(i)$$

Here we assume that it is clear that average payoffs $\overline{\pi}_C(p)$ and $\overline{\pi}_D(p)$ depend on the frequency $p$ of co-operators in the overall population, whereas game payoffs $\pi_C(i)$ and $\pi_D(i)$ depend on the number of co-operators in the group. An equilibrium is stable, if co-operators have a higher average payoff than defectors do for $p<p^*$, and defectors have a higher payoff than co-operators for $p>p^*$.

At $p = 0$, all defectors find themselves in groups with no co-operators, and co-operators cannot invade if their average payoff is lower than that of defectors. This gave us the criterion that we used before:

$$\lim_{p\downarrow 0}\sum_{i=0}^{n} u_{i,C}(p)\pi_C(i) = \sum_{i=0}^{n}\underline{u}_i\pi_C(i) < \pi_D(0)$$

At $p = 1$, all co-operators find themselves in groups with no defectors, and defectors cannot invade if their average payoff is lower than that of co-operators. This gave us the criterion that we used before:

$$\pi_C(i) > \sum_{i=0}^{n}\overline{u}_i\pi_D(i) = \lim_{p\uparrow 1}\sum_{i=0}^{n} u_{i,D}(p)\pi_D(i)$$

## C. Costs and benefits

The costs and benefits an individual faces may depend on the type of group an individual finds itself in. The composition of the population $f_i(p)$, indicating what the frequencies of the different types of groups will be, depends on the overall frequency $p$ of co-operators. The average costs and benefits therefore may also depend on this frequency $p$. The counterfactual

method, as originally suggested by *Karlin and Matessi (1983)*, *Matessi and Karlin (1984)* and *Matessi and Karlin (1986)* only considers the costs actually bore by co-operators, and the benefits actually conferred by them onto others. In Section 3 of *van Veelen et al. (2017)* we show that the costs and benefits of cooperation (compared to defection) then are not necessarily minus the costs and benefits of defection (compared to cooperation). We therefore suggested to include all individuals that faced the opportunity to cooperate, whether they ended up cooperating or defecting. This definition is consistent, and this is also the definition we use here. The cost for a co-operator – comparing its payoffs when it cooperates to its payoffs had it defected – then is $\pi_D(i-1) - \pi_C(i)$; the cost for a defector is $\pi_D(i) - \pi_C(i+1)$. Average costs then become

$$c(p) = \sum_{i=0}^{n} f_i(p) \left[ \frac{i}{n}(\pi_D(i-1) - \pi_C(i)) + \frac{n-i}{n}(\pi_D(i) - \pi_C(i+1)) \right]$$

Aggregate average benefits similarly become

$$b(p) = \sum_{i=0}^{n} f_i(p) \left[ \frac{i}{n}\{(i-1)(\pi_C(i) - \pi_C(i-1))) + (n-i)(\pi_D(i) - \pi_D(i-1))\} \right.$$
$$\left. + \frac{n-i}{n}\{(i)(\pi_C(i+1) - \pi_C(i)) + (n-i-1)(\pi_D(i+1) - \pi_D(i))\} \right]$$

## D. Comparison

Previously, in Section V, we have seen that inclusive fitness $r(p)b(p) - c(p)$ may have a sign that is different from the sign of $\overline{\pi}_C(p) - \overline{\pi}_D(p)$, at $p = 0$ and/or at $p = 1$, even though that may not always be observable in equilibrium. In a mixed equilibrium $p^*$, where $\overline{\pi}_C(p^*) - \overline{\pi}_D(p^*) = 0$, there is also no reason why $r(p^*)b(p^*) - c(p^*)$ would be 0 too – unless there is equal gains from switching. Here this discrepancy would be observable (see *Appendix 1—figure 5*). Note that equal gains would be at odds with there being a stable mixed equilibrium at 0<$p^*$<1. Details of the computations, as well as the computation of costs and benefits for the regression method, can be found in Section IX. With equal gains, $\pi_C(i) - \pi_C(i-1) = \pi_D(j) - \pi_D(j-1))$ for all $i,j \in \{1, ..., n\}$ and $\pi_D(i-1) - \pi_C(i)) = \pi_D(j-1) - \pi_C(j))$ for all $i,j \in \{1, ..., n\}$. If we denote these constants by $\mathbf{b}$ and $\mathbf{c}$, then inclusive fitness becomes $r(p)(n-1)\mathbf{b} - \mathbf{c}$ – where $(n-1)\mathbf{b}$ is the aggregate benefits. This is found by filling in the constants in the formulas at subsection C above.

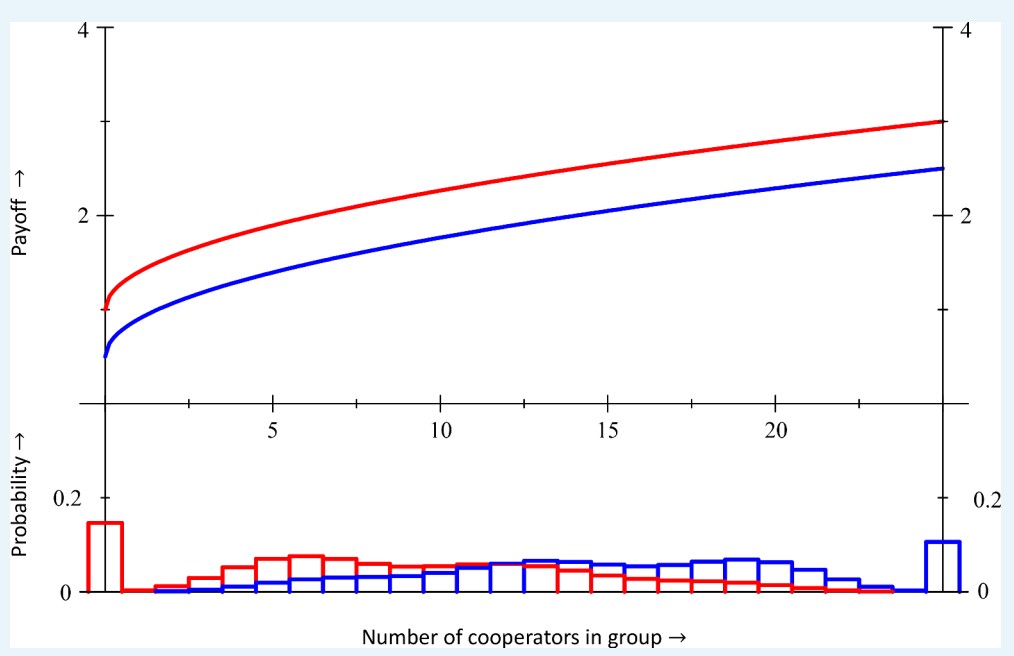

**Appendix 1—figure 5.** The previous figures depicted $\underline{u}$, which is the population structure profile at $p = 0$, and $\overline{u}$, which is the population structure profile at $p = 1$. The payoffs co-operators get, when they find themselves in groups according to $\underline{u}$ would be compared to the payoffs defectors get when they are in a group of defectors only in order to determine whether co-operators can invade defectors, at $p = 0$, and the mirror image of that in order to determine whether defectors can invade co-operators, at $p = 1$. The $u_{i,C}(p^*)$ and $u_{i,D}(p^*)$ depicted here all pertain to one and the same equilibrium $p^*$, at which $\overline{\pi}_C(p^*) - \overline{\pi}_D(p^*) = 0$. With the same payoff function as at **Appendix 1—figure 4**, we find an equilibrium frequency of co-operators of $p^* = 0.473$, at which inclusive fitness is $\frac{1}{4}(2.103) - (0.412) \approx 0.113 \neq 0$.

DOI: https://doi.org/10.7554/eLife.41901.012

### E. If $\pi_D$ and $\pi_C$ are linear, but not parallel, they still violate 'equal gains from switching'

Equal gains from switching means that the effect of any given individual changing from defection to cooperation on the fitness of other individuals and on her own fitness is independent of what these other individuals choose. The examples of violations above, depicted in **Appendix 1—figures 3–7**, all have non-linear $\pi_D$ and $\pi_C$. Violations can also occur when $\pi_D$ and $\pi_C$ are linear, but not parallel – in which case the fitness effects depend on whether the recipient is a co-operator or defector itself.

An example would be $\pi_D = 1 + 3\frac{i}{n}$ and $\pi_C = 2 + \frac{i}{n}$. Both of these are linear in the number of co-operators, but they are not equally steep. Switching to cooperation benefits the other more if the other is a defector, compared to if the other is a co-operator. The total benefit one can confer to others therefore is higher in a group in which most of the others are defectors, and it is lower in a group in which most of the other group members are co-operators. If we choose the same population structure as for **Appendix 1—figures 2–5**, then there is a mixed equilibrium – at $p = \frac{8}{9}$ – at which Hamilton's rule is violated; $\frac{1}{4}\left(\frac{88}{75}\right) - \left(\frac{2}{3}\right) \approx 0.373 \neq 0$.

### F. An aside about the regression method

It might also be good to illustrate how the regression method would have treated this fitness function. The regression method chooses $\alpha$, $\beta$ and $\gamma$ so as to minimize

$$\sum_{i=0}^{n} f_i(p)\left(\frac{i}{n}\left(\pi_C(i) - \left(\alpha + \beta\frac{i-1}{n-1} - \gamma\right)\right)^2 + \frac{n-i}{n}\left(\pi_D(i) - \left(\alpha + \beta\frac{i}{n-1}\right)\right)^2\right),$$

where $\alpha$ is the intercept, $\beta$ is interpreted as $n-1$ times the benefit of being matched with one additional co-operator and $\gamma$ is interpreted as the cost of being a co-operator oneself. It is easier to illustrate this without population structure, so the figures on the next page show what this would result in for the fitness function of the last example, combined with the well mixed population, and a group size of $n = 25$. In Section IX we will also consider a case with population structure.

The $\alpha$, $\beta$ and $\gamma$ are computed in the limit of $p \downarrow 0$ (top panel), at $p = \frac{1}{2}$ (middle panel), and in the limit of $p \uparrow 1$ (bottom panel). For the well mixed population, it is not too hard to compute these in general at $p \downarrow 0$ and $p \uparrow 1$. In the limit of $p \downarrow 0$, the regression method always results in $\alpha = \pi_D(0)$. The reason is that $\lim_{p\downarrow 0} f_0(p) = 1$, and $\lim_{p\downarrow 0} f_i(p) = 0$ for all $i > 0$. Therefore, in the limit, for sure $(\pi_D(0) - \alpha)^2$ must be minimized. Because $f_1(p)$ is of the order $p$, while $f_i(p)$ is of the order $p^2$ or higher, for $i > 1$, $\beta$ and $\gamma$ must, in the limit, also minimize $\frac{1}{n}(\pi_C(1) - (\alpha - \gamma))^2 + \frac{n-1}{n}\left(\pi_D(1) - \left(\alpha + \beta\frac{1}{n-1}\right)\right)^2$. This can be done here by simply choosing $\gamma = \alpha - \pi_C(1) = \pi_D(0) - \pi_C(1)$, and $\beta = (n-1)(\pi_D(1) - \alpha) = (n-1)(\pi_D(1) - \pi_D(0))$. The 'estimated' $\pi_D$ at $p \downarrow 0$ therefore is always a straight line, the steepness of which derives from the first step, from $i = 0$ to $i = 1$, in $\pi_D(i)$. The 'estimated' $\pi_C$ is a line parallel to $\pi_D$, but then shifted up by $\pi_D(0) - \pi_C(1)$, and to the right by $\frac{1}{n-1}$.

In this particular case, the squared difference at $p = \frac{1}{2}$ is minimized by choosing $\gamma = 0$, making the 'estimated' $\pi_C$ and $\pi_D$ coincide. With population structure, this will all be a bit different, but what remains true is that the 'estimated' $\pi_C$ and $\pi_D$ will always be parallel straight lines, whether the true $\pi_C$ and $\pi_D$ are straight and parallel or not. The slope and positions of these straight parallel lines typically changes with $p$. The reason why this particular choice for (mis)specifying $\pi_C$ and $\pi_D$ would lead to Hamilton's rule always holding, is given in Section 4 of **van Veelen et al. (2017)**.

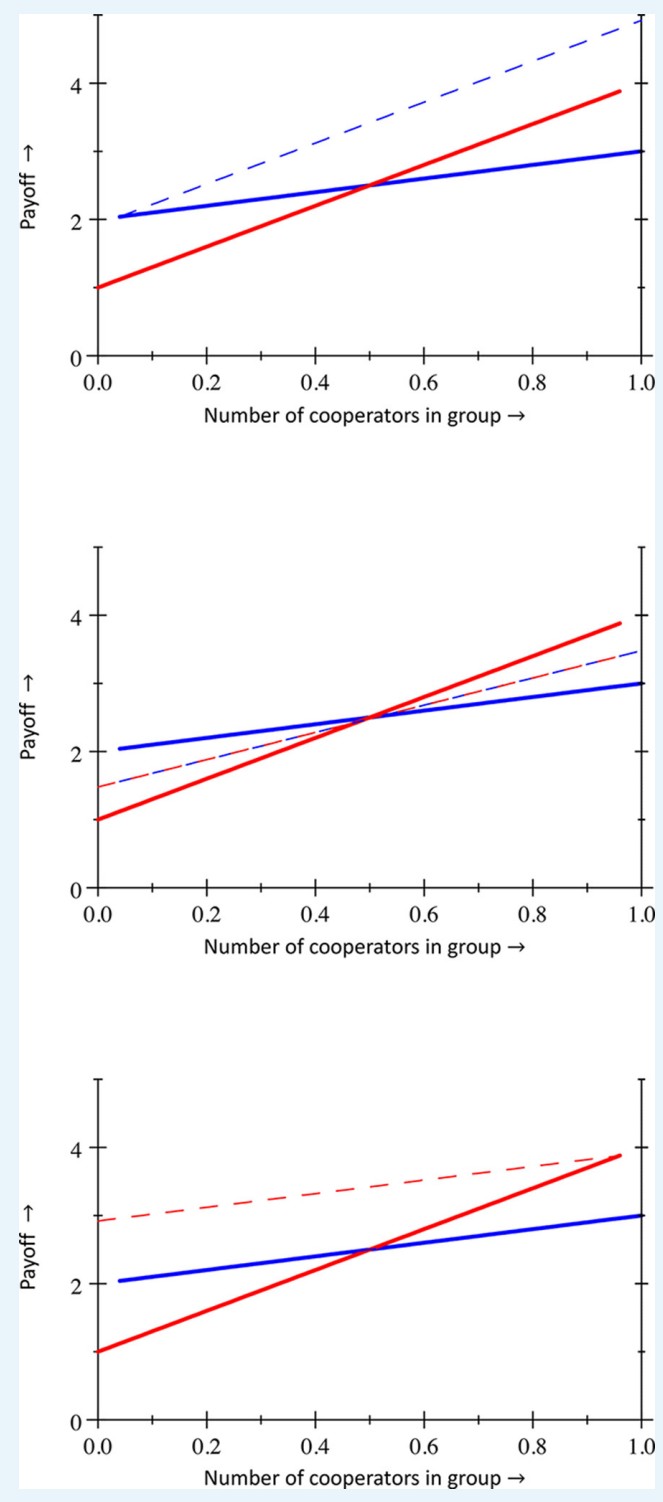

**Appendix 1—figure 6.** The $\pi_D$ according to the regression method coincides the true $\pi_D$ at $p \downarrow 0$ (top panel, red), but the $\pi_C$ according to the regression method (dotted blue) differs from the true $\pi_C$ (solid blue). The $\pi_D$ and $\pi_C$ according to the regression method coincide with each other at $p = 0.5$ (middle panel, dotted lines) but do not coincide with the true $\pi_D$ (solid red) and $\pi_C$ (solid blue). The $\pi_C$ according to the regression method coincides the true $\pi_C$ at $p \uparrow 1$ (bottom panel, blue), but the $\pi_D$ according to the regression method (dotted red) differs from the true $\pi_D$ (solid red).

DOI: https://doi.org/10.7554/eLife.41901.013

# VIII. Continuous frequencies

For proper large group sizes it can be easier, and more natural, to work with continuous distributions of within-group frequencies, instead of functions $f_i(p)$, which are only defined for $i = 0, ..., n$. In this section we will therefore introduce the continuous counterparts of definitions from previous sections.

## A. Relatedness at the edges

Let $\{F_p\}_{p \in [0,1]}$ be a set of cumulative distribution functions $F_p : [0, 1] \to [0, 1]$, one for every frequency $p$ of co-operators in the population as a whole. $F_p(x)$ is the probability, at frequency $p$, that the share of co-operators in a randomly chosen group is smaller than or equal to $x$. Consistency requires that $\int_0^1 x dF_p(x) = p$, and therefore that $F_p$ converges weakly to the Dirac measure at 0 for $p \downarrow 0$, and to the Dirac measure at 1 as $p \uparrow 1$. If $F_p$ has a density, then one can write $\int_0^1 g(x) dF_p(x)$ as $\int_0^1 g(x) f_p(x) dx$.

To relate this to the setting in previous sections, with discrete within-group frequencies, one would normalize those group types to be values in $[0, 1]$, and let them be $\{0, \frac{1}{n}, ..., 1\}$ instead of $\{0, ..., n\}$. Furthermore, one would turn a population structure characterized by functions $f_i(p)$, $i = 0, ..., n$, into one characterized by a cumulative distributions by choosing $F_p(x) = \sum_{j=0}^{i} f_j(p)$ if $\frac{i-1}{n} < x \le \frac{i}{n}$. For cumulative distributions $F_p$ with a density $f_p$, this density is the natural counterpart of the vector with $f_i(p)$ as its $i$th element. The notation is a bit different, though; the overall frequency $p$ is a subscript in the one, and an argument in the other case, while group composition $x$ is an argument in the one, and its counterpart $i$ is a subscript in the other.

Now define cumulative distribution functions $\underline{U}$ and $\overline{U}$ as follows:

$$\underline{U}(x) = \lim_{p \downarrow 0} \int_0^x \frac{y dF_p(y)}{p}$$

and

$$\overline{U}(x) = \lim_{p \uparrow 1} \int_0^x \frac{y dF_p(y)}{p}$$

and assume that these limits exist.

Again, this one could relate to earlier sections; we can get the vectors $\underline{u}$ and $\overline{u}$ back by simply 'uncumulating' $\underline{U}$ and $\overline{U}$; define $\underline{u}_i = \underline{U}(\frac{i}{n}) - \underline{U}(\frac{i-1}{n})$ and $\overline{u}_i = \overline{U}(\frac{i}{n}) - \overline{U}(\frac{i-1}{n})$.

The fitness functions here will also be defined on $[0, 1]$. They will be denoted by $\pi_C(x)$ for the fitness of a co-operator in a group with frequency of co-operators $x$, and $\pi_D(x)$ for its defector counterpart. Analogous to the derivation in Section II, relatedness at $p = 0$ is $\int_0^1 x d\underline{U}(x)$, and at $p = 1$ it is $\int_0^1 (1 - x) d\overline{U}(x)$. Moreover, it would be natural to assume that $\overline{U}(x) = \underline{U}(1 - x)$, and therefore that relatedness at 0 and 1 are equal.

## B. Invasions and inclusive fitness

### At $p = 0$

The more general version of *Equation 1* is that cooperation cannot invade at $p = 0$ if their average payoff is less than that of defectors:

$$\int_0^1 \pi_C(x) d\underline{U}(x) < \pi_D(0) \tag{7}$$

Aggregate benefits according to the counterfactual method are defined analogously to Section IV. There they were defined at $p = 0$ as $n - 1$ times the difference between the payoff of a defector that finds itself in a group with one co-operator and defector that finds itself in a

group with zero co-operators. In normalized terms, that would be $b(0) = (n-1)\left[\pi_D\left(\frac{1}{n}\right) - \pi_D(0)\right]$, which, if $n$ were to go to infinity, would amount to $b(0) = \frac{d_+\pi_D}{dx}(0)$, where $\frac{d_+\pi_D}{dx}(0)$ is the right derivative of $\pi_D$ at 0. Similarly at $p = 1$, where aggregate benefits are $b(1) = \frac{d_-\pi_C}{dx}(1)$, which is the left derivative of $\pi_C$ at 1. (Discrete population structures, with finite group size, are subsumed under the more general case with distribution functions. Computation of costs and benefits there, however, do differ from how they are computed here, because here one individual changing strategies has no effect on within group frequency $x$, while in the finite case that is not true.)

With those benefits, we can write the continuous version of **Equation 2**. At $p = 0$ inclusive fitness is negative if

$$\int_0^1 x d\underline{U}(x) \times \frac{d_+\pi_D}{dx}(0) < \pi_D(0) - \pi_C(0)$$

which can be rewritten as

$$\int_0^1 \left(\pi_C(0) + x\frac{d_+\pi_D}{dx}(0)\right) d\underline{U}(x) < \pi_D(0) \tag{8}$$

If $\pi_C(0) + x\frac{d_+\pi_D}{dx}(0) = \pi_C(x)$, then the left hand sides of **Equations 7 and 8** are the same, and inclusive fitness will always agree with the direction of selection at $p = 0$. If $\pi_C(0) + x\frac{d_+\pi_D}{dx}(0) < \pi_C(x)$ the left hand sides of **Equations 7 and 8** will not be the same, but a violation in equilibrium at $p = 0$ will not be observed; if inequality **Equation 7** holds, and co-operators cannot invade, then certainly **Equation 8** will hold, making inclusive fitness negative, as its left hand side is even smaller than the left hand side of **Equation 7**.

## At $p = 1$

Similarly, defection cannot invade at $p = 1$ if

$$\int_0^1 \pi_D(x) d\overline{U}(x) < \pi_C(1) \tag{9}$$

while inclusive fitness at $p = 1$ is positive if

$$\int_0^1 (1-x) d\overline{U}(x) \times \frac{d_-\pi_C}{dx}(1) > \pi_D(1) - \pi_C(1)$$

which can be rewritten as

$$\int_0^1 \left(\pi_D(1) - (1-x)\frac{d_-\pi_C}{dx}(1)\right) d\underline{U}(x) < \pi_C(1) \tag{10}$$

If $\pi_D(1) - (1-x)\frac{d_-\pi_C}{dx}(1) = \pi_D(x)$, then the left hand sides of **Equations 9 and 10** are the same. If $\pi_D(1) - (1-x)\frac{d_-\pi_C}{dx}(1) < \pi_D(x)$, the left hand sides of **Equations 9 and 10** will not be the same, but a violation in equilibrium at $p = 1$ will not be observed; if inequality **Equation 9** holds, and defectors cannot invade, then certainly **Equation 10** will hold, making inclusive fitness positive, as its left hand side is even smaller than the left hand side of **Equation 9**.

## C. Mixed equilibria

Again, when co-operators can invade defectors and defectors can invade co-operators, then for sure there must be at least one equilibrium mixture of co-operators and defectors. Also other games without equal gains from switching may have stable frequencies $p$ at which co-operators and defectors coexist. At those frequencies, we can again compare the direction of selection with inclusive fitness. Here we will assume that at the equilibrium frequency $p^*$, $F_{p^*}(x)$ has a density $f_p^*(x)$, which allows us to write $f_{p^*}(x)dx$ for $dF_{p^*}(x)$.

## Equilibrium

An equilibrium is a frequency $p^*$ for which the average payoff of co-operators and defectors are equal; $\overline{\pi}_C(p^*) = \overline{\pi}_D(p^*)$, that is,

$$\frac{\int_0^1 x f_{p^*}(x) \pi_C(x) dx}{p^*} = \frac{\int_0^1 (1-x) f_{p^*}(x) \pi_D(x) dx}{1-p^*}$$

## Inclusive Fitness

Relatedness, benefits and costs are computed as before, in Section VII, but now with a continuous $f_p$.

$$r(p) = \frac{\int_0^1 x^2 f_p(x) dx}{p} - \frac{\int_0^1 x(1-x) f_p(x) dx}{1-p}$$

$$b(p) = \int_0^1 \left( x \frac{d}{dx} \pi_C(x) + (1-x) \frac{d}{dx} \pi_D(x) \right) f_p(x) dx$$

$$c(p) = \int_0^1 (\pi_D(x) - \pi_C(x)) f_p(x) dx$$

Again, there is no reason why in equilibrium, where $\overline{\pi}_C(p^*) - \overline{\pi}_D(p^*) = 0$, also $r(p^*)b(p^*) - c(p^*)$ would be equal to 0.

## Example

Consider the following payoff function

$$\begin{aligned}\pi_C(x) &= 1 + \beta x^\alpha - \gamma \\ \pi_D(x) &= 1 + \beta x^\alpha\end{aligned}$$

Then $p^*$ is an equilibrium frequency if

$$\beta \left( \frac{\int_0^1 x^{1+\alpha} f_{p^*}(x) dx}{p^*} - \frac{\int_0^1 (1-x) x^\alpha f_{p^*}(x) dx}{1-p^*} \right) - \gamma = 0$$

Inclusive fitness is

$$\beta \int_0^1 \alpha x^{\alpha-1} f_p(x) dx \left( \frac{\int_0^1 x^2 f_p(x) dx}{p} - \frac{\int_0^1 (1-x) x f_p(x) dx}{1-p} \right) - \gamma$$

and unless $\alpha = 1$, the last expression is typically not 0 for $p = p^*$.

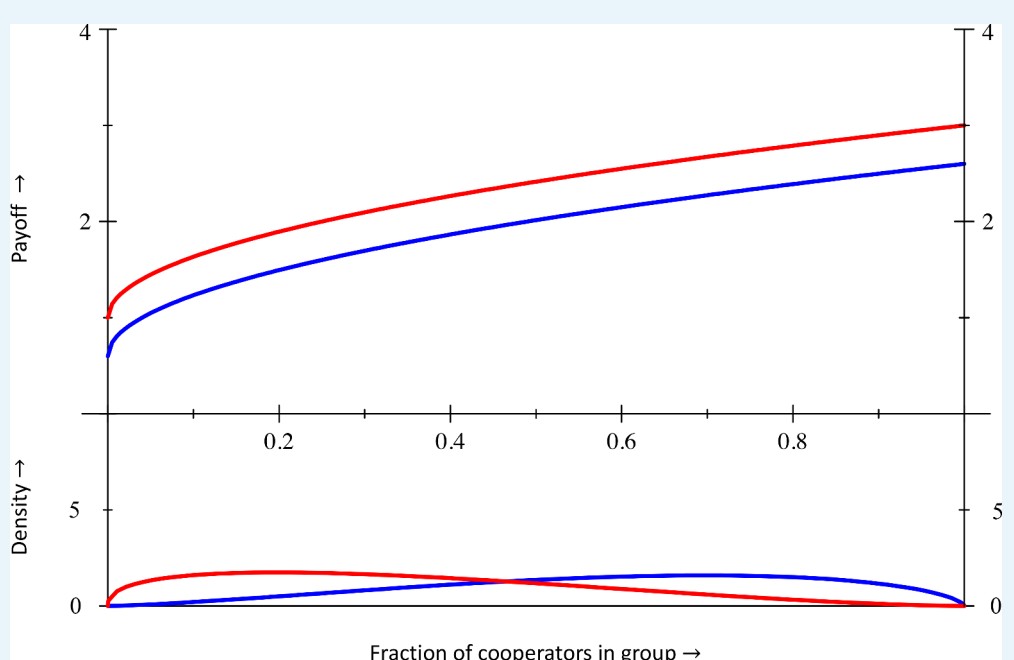

**Appendix 1—figure 7.** With parameters $\alpha = \frac{1}{2}$, $\beta = 2$ and $\gamma = 0.4$, the equilibrium frequency of co-operators, at which $\overline{\pi}_C(p^*) = \overline{\pi}_D(p^*)$, is $p^* = 0.465$. Inclusive fitness there is $\frac{1}{4}(1.82) - (0.4) \approx 0.055 \neq 0$. Section X specifies the $f_p(x)$ for this example.

DOI: https://doi.org/10.7554/eLife.41901.014

## IX. Discrete population structures

The population structure used in the examples is not meant to be realistic or reflect any particular real population structure; it is just a convenient vehicle to generate a variety of structures. Imagine a parent group that is assembled by randomly drawing $m$ parents from a large population, a share $p$ of which are co-operators. A parent group consisting of $k$ co-operators and $m - k$ defectors is drawn with probability

$$\binom{m}{k}(1-p)^{m-k}p^k$$

Then an offspring group is made by drawing – with replacement – $n$ times a parent who reproduces. The probability that a parent group with $k$ co-operators reproduces an offspring group with $i$ co-operators is

$$q_i(k) = \binom{n}{i}\left(\frac{k}{m}\right)^i\left(\frac{m-k}{m}\right)^{n-i}$$

This then implies the following population structure, with

$$f_i(p) = \sum_{k=0}^{m}\binom{m}{k}(1-p)^{m-k}p^k \cdot q_i(k)$$

for $i = 0, ..., n$. This is a consistent population structure. Firstly, the frequencies of different group types add up to one.

$$\sum_{i=0}^{n} f_i(p) = \sum_{i=0}^{n} \sum_{k=0}^{m} \binom{m}{k}(1-p)^{m-k}p^k \cdot q_i(k)$$

$$= \sum_{k=0}^{m} \binom{m}{k}(1-p)^{m-k}p^k \sum_{i=0}^{n} \binom{n}{i}\left(\frac{k}{m}\right)^i \left(\frac{m-k}{m}\right)^{n-i}$$

$$= \sum_{k=0}^{m} \binom{m}{k}(1-p)^{m-k}p^k = 1$$

Here we use twice that the probabilities of the binomial distribution add up to 1. Also, the frequency of co-operators in the offspring population is consistent with $p$.

$$\sum_{i=0}^{n} \frac{i}{n} f_i(p) = \sum_{i=0}^{n} \frac{i}{n} \sum_{k=0}^{m} \binom{m}{k}(1-p)^{m-k}p^k \cdot q_i(k)$$

$$= \sum_{k=0}^{m} \binom{m}{k}(1-p)^{m-k}p^k \sum_{i=0}^{n} \frac{i}{n}\binom{n}{i}\left(\frac{k}{m}\right)^i \left(\frac{m-k}{m}\right)^{n-i}$$

$$= \sum_{k=0}^{m} \binom{m}{k}(1-p)^{m-k}p^k \left(\frac{k}{m}\right) \sum_{i=1}^{n} \binom{n-1}{i-1}\left(\frac{k}{m}\right)^{i-1} \left(\frac{m-k}{m}\right)^{n-i}$$

$$= p\sum_{k=1}^{m} \binom{m-1}{k-1}(1-p)^{m-k}p^{k-1} = p$$

Again we twice use that the probabilities of the binomial distribution add up to 1.

Relatedness is

$$r(p) = P(C|C) - P(C|D)$$

We first compute $P(C|C)$ in a straightforward, but somewhat elaborate way.

$$P(C|C) = \frac{1}{p}\sum_{i=2}^{n} f_i(p)\frac{i}{n}\frac{i-1}{n-1} = \frac{1}{p}\sum_{i=2}^{n} \sum_{k=0}^{m} \binom{m}{k}(1-p)^{m-k}p^k \cdot \binom{n}{i}\left(\frac{k}{m}\right)^i \left(\frac{m-k}{m}\right)^{n-i} \cdot \frac{i}{n}\frac{i-1}{n-1}$$

$$= \frac{1}{p}\sum_{k=0}^{m} \binom{m}{k}(1-p)^{m-k}p^k \cdot \sum_{i=2}^{n} \binom{n-2}{i-2}\left(\frac{k}{m}\right)^i \left(\frac{m-k}{m}\right)^{n-i}$$

$$= \frac{1}{p}\sum_{k=0}^{m} \binom{m}{k}(1-p)^{m-k}p^k \cdot \left(\frac{k}{m}\right)^2 \sum_{i=2}^{n} \binom{n-2}{i-2}\left(\frac{k}{m}\right)^{i-2} \left(\frac{m-k}{m}\right)^{n-i}$$

$$= \frac{1}{p}\sum_{k=0}^{m} \binom{m}{k}(1-p)^{m-k}p^k \cdot \left(\frac{k}{m}\right)^2 = \frac{1}{m}\sum_{k=1}^{m} k\binom{m-1}{k-1}(1-p)^{m-k}p^{k-1}$$

$$= \frac{1}{m}\sum_{k=1}^{m} (1+(k-1))\binom{m-1}{k-1}(1-p)^{m-k}p^{k-1} = \frac{1}{m}(1+(m-1)p) = \frac{1}{m} + \left(1-\frac{1}{m}\right)p$$

Here we used twice that the probabilities in the binomial distribution add up to 1, again, and once that the expectation of the binomial distribution, with $m-1$ trials and probability $p$, is $(m-1)p$.

This answer makes perfect sense, because if an individual is a $C$ itself, then its parent is one too, while the other $m-1$ individuals in the parent group are random draws, with probability $p$ of also being a $C$. The probability that a randomly drawn other offspring is also a $C$ therefore is $\frac{1}{m} + \left(1-\frac{1}{m}\right)p$. Similarly, one can compute $P(C|D)$ to be $\left(1-\frac{1}{m}\right)p$, which makes $r(p) = P(C|C) - P(C|D) = \frac{1}{m}$, which is actually independent of $p$.

The population structure profile at $p=0$ for a population structure from this family is $\underline{u}_0 = 0$ and

$$\underline{u}_i = \lim_{p\downarrow 0} \frac{\frac{i}{n}f_i(p)}{p} = \binom{n-1}{i-1}\left(\frac{1}{m}\right)^{i-1}\left(\frac{m-1}{m}\right)^{n-i}, \quad i=1,...,n$$

This can be found by simply evaluating the limits. The easier intuition is that in the limit of $p \downarrow 0$, all co-operator parents find themselves in parent groups in which they are the only one,

and all others are defector parents. This means that, conditional on being a co-operator offspring, the composition of the remainder of its offspring group follows a binomial distribution, with the probability of any draw being a co-operator equal to $\frac{1}{m}$.

Similarly, the population structure profile at $p = 1$ for a population structure from this family is $\underline{u}_n = 0$ and

$$\bar{u}_i = \lim_{p\uparrow 1} \frac{\frac{n-i}{n} f_i(p)}{1-p} = \binom{n-1}{i} \left(\frac{m-1}{m}\right)^i \left(\frac{1}{m}\right)^{n-i-1}, \quad i = 0, ..., n-1$$

Now, conditional on being a defector offspring, the composition of the remainder of the offspring group it is in follows a binomial distribution, with the probability of any draw being a co-operator equal to $\frac{m-1}{m}$.

The population structures used for **Appendix 1—figures 1–5** all have $n = 25$ for the size of the offspring group. **Appendix 1—figures 1a** has $m = 10$ for the size of the parent group, and hence relatedness $\frac{1}{10}$, as does **Figure 2a** in the main text. **Appendix 1—figures 1b through 5** here, and their corresponding figures in the main text, feature a parent group size of $m = 4$, and hence relatedness $\frac{1}{4}$. All computations of $\overline{\pi}_C(p) - \overline{\pi}_D(p)$ and all computations of inclusive fitness are done by combining these population structures with the different payoff functions from Sections III, IV and V, as described below. Mathematica files with further computational details can be obtained from the author.

### Figure 2a in the main text

At $p = 0$, and with $m = 10$, $\sum_{i=1}^{25} \underline{u}_i \pi_C(i) = \sum_{i=1}^{25} \binom{25-1}{i-1} \left(\frac{1}{m}\right)^{i-1} \left(\frac{m-1}{m}\right)^{25-i} \cdot \left(0.6 + 2\left(\frac{i}{25}\right)\right) = 0.872$,

while $\pi_D(0) = 1 + 2\left(\frac{0}{25}\right) = 1$, and therefore $\sum_{i=1}^{25} \underline{u}_i \pi_C(i) - \pi_D(0) = -0.128$. According to the counterfactual method, benefits are $(n-1)(\pi_D(1) - \pi_D(0)) = 24\left(1 + 2\left(\frac{1}{25}\right) - \left(1 + 2\left(\frac{0}{25}\right)\right)\right) = \frac{48}{25}$, costs are $\pi_D(0) - \pi_C(1) = 1 + 2\left(\frac{0}{25}\right) - \left(0.6 + 2\left(\frac{1}{25}\right)\right) = \frac{8}{25}$ and inclusive fitness is $\frac{1}{10}\frac{48}{25} - \frac{8}{25} = -0.128$.

At $p = 1$, and with $m = 10$, $\pi_C(25) = 0.6 + 2\left(\frac{25}{25}\right) = 2.6$, while

$\sum_{i=0}^{25-1} \bar{u}_i \pi_D(i) = \sum_{i=0}^{25-1} \binom{25-1}{i} \left(\frac{m-1}{m}\right)^i \left(\frac{1}{m}\right)^{25-i-1} \cdot \left(1 + 2\left(\frac{i}{25}\right)\right) = 2.728$, and therefore

$\pi_C(25) - \sum_{i=0}^{25-1} \bar{u}_i \pi_D(i) = -0.128$. According to the counterfactual method, benefits are $(n-1)(\pi_C(25) - \pi_C(24)) = 24\left(0.6 + 2\left(\frac{25}{25}\right) - \left(0.6 + 2\left(\frac{24}{25}\right)\right)\right) = \frac{48}{25}$, costs are $\pi_D(24) - \pi_C(25) = 1 + 2\left(\frac{24}{25}\right) - \left(0.6 + 2\left(\frac{25}{25}\right)\right) = \frac{8}{25}$, and inclusive fitness is $\frac{1}{10}\frac{48}{25} - \frac{8}{25} = \frac{4}{25} = -0.128$.

Because the true fitness function is linear, the regression method would give the same costs and benefits, both at $p = 0$ and at $p = 1$.

### Appendix 1—figure 2 (Figure 2b in the main text)

At $p = 0$, and with $m = 4$, $\sum_{i=1}^{25} \underline{u}_i \pi_C(i) = \sum_{i=1}^{25} \binom{25-1}{i-1} \left(\frac{1}{m}\right)^{i-1} \left(\frac{m-1}{m}\right)^{25-i} \cdot \left(0.6 + 2\left(\frac{i}{25}\right)\right) = 1.16$,

while $\pi_D(0) = 1 + 2\left(\frac{0}{25}\right) = 1$, and therefore $\sum_{i=1}^{25} \underline{u}_i \pi_C(i) - \pi_D(0) = 0.16$. Benefits and costs according to the counterfactual method are the same as in **Figure 2a** in the main text, which makes inclusive fitness equal to $\frac{1}{4}\frac{48}{25} - \frac{8}{25} = \frac{4}{25} = 0.16$.

At $p = 1$, and with $m = 4$, $\pi_C(25) = 0.6 + 2\left(\frac{25}{25}\right) = 2.6$, while

$\sum_{i=0}^{25-1} \bar{u}_i \pi_D(i) = \sum_{i=0}^{25-1} \binom{25-1}{i} \left(\frac{m-1}{m}\right)^i \left(\frac{1}{m}\right)^{25-i-1} \cdot \left(1 + 2\left(\frac{i}{25}\right)\right) = 2.44$, and therefore

$\pi_C(25) - \sum_{i=0}^{25-1} \bar{u}_i \pi_D(i) = 0.16$. Benefits and costs according to the counterfactual method are the same as in **Figure 2a** in the main text, which makes inclusive fitness equal to $\frac{1}{4}\frac{48}{25} - \frac{8}{25} = \frac{4}{25} = 0.16$.

Because the true fitness function is linear, the regression method would give the same costs and benefits, both at $p = 0$ and at $p = 1$.

### Appendix 1—figure 3 (Figure 2c in the main text)

At $p = 0$, $\sum_{i=1}^{25} \underline{u}_i \pi_C(i) = \sum_{i=1}^{25} \binom{25-1}{i-1} \left(\frac{1}{m}\right)^{i-1} \left(\frac{m-1}{m}\right)^{25-i} \cdot \left(0.5 + 2\left(\frac{i}{25}\right)^2\right) = 0.6712$, while

$\pi_D(0) = 1 + 2\left(\frac{0}{25}\right)^2 = 1$, and therefore $\sum_{i=1}^{25} \underline{u}_i \pi_C(i) - \pi_D(0) = -0.3288$.

According to the **counterfactual method**, benefits at $p = 0$ are

$(n-1)(\pi_D(1) - \pi_D(0)) = 24\left(1 + 2\left(\frac{1}{25}\right)^2 - \left(1 + 2\left(\frac{0}{25}\right)^2\right)\right) = \frac{48}{625}$. Costs at $p = 0$ are

$\pi_D(0) - \pi_C(1) = 1 + 2\left(\frac{0}{25}\right)^2 - \left(0.5 + 2\left(\frac{1}{25}\right)^2\right) = \frac{1}{2} - \frac{2}{625}$ and inclusive fitness is

$\frac{1}{4}\frac{48}{625} - \frac{1}{2} + \frac{2}{625} = -0.4776$.

In the limit of $p \downarrow 0$, all groups are groups with defectors only. The **regression method** therefore would imply that $\alpha = \pi_D(0) = 1$ (see also Section VII F). Furthermore, minimizing the squared difference between true fitness's and the fitness's according to a linear model, given the relative weights of all frequencies of groups with $i$ co-operators, with $i \geq 1$, would then amount to

$$\min_{\beta, \gamma} \sum_{i=1}^{n} \underline{u}_i \left[ \left(\pi_C(i) - \left(1 + \beta\left(\frac{i-1}{n-1}\right) - \gamma\right)\right)^2 + \frac{n-i}{i}\left(\pi_D(i) - \left(1 + \beta\left(\frac{i}{n-1}\right)\right)\right)^2 \right]$$

resulting in $b = \beta^* = 0.586971$, and $c = \gamma^* = 0.475543$. This makes inclusive fitness $rb - c = -0.3288$.

At $p = 1$, $\pi_C(25) = 0.5 + 2\left(\frac{25}{25}\right)^2 = 2.5$, while

$\sum_{i=0}^{25-1} \bar{u}_i \pi_D(i) = \sum_{i=0}^{25-1} \binom{25-1}{i} \left(\frac{m-1}{m}\right)^i \left(\frac{1}{m}\right)^{25-i-1} \cdot \left(1 + 2\left(\frac{i}{25}\right)^2\right) = 2.0512$, and therefore

$\pi_C(25) - \sum_{i=0}^{25-1} \bar{u}_i \pi_D(i) = 0.4488$.

According to the **counterfactual method**, benefits at $p = 1$ are

$(n-1)(\pi_C(25) - \pi_C(24)) = 24\left(0.5 + 2\left(\frac{25}{25}\right)^2 - \left(0.5 + 2\left(\frac{24}{25}\right)^2\right)\right) = \frac{2352}{625} = 3.7632$. Costs at $p = 1$ are

$\pi_D(24) - \pi_C(25) = 1 + 2\left(\frac{24}{25}\right)^2 - \left(0.5 + 2\left(\frac{25}{25}\right)^2\right) = \frac{1}{2} - \frac{98}{625}$, and inclusive fitness is

$\frac{1}{4}\frac{2352}{625} - \frac{1}{2} + \frac{98}{625} = 0.5976$.

Because in the limit of $p \uparrow 1$, all groups are groups with co-operators only, the **regression method** would imply that $\alpha + \beta - \gamma = \pi_C(25) = 2.5$ (see also Section VII F). Minimizing the squared difference between true fitness's and the fitness's according to a linear model, considering all frequencies of groups with less than $n$ co-operators, would then amount to

$$\min_{\beta, \gamma} \sum_{i=0}^{n-1} \bar{u}_i \left[ \frac{i}{n-i}\left(\pi_C(i) - \left(2.5 - \beta\left(\frac{n-i}{n-1}\right)\right)\right)^2 + \left(\pi_D(i) - \left(2.5 - \beta\left(\frac{n-i-1}{n-1}\right) + \gamma\right)\right)^2 \right]$$

resulting in $b = \beta^* = 3.25303$, and $c = \gamma^* = 0.364457$. This makes inclusive fitness $rb - c = 0.4488$.

### Appendix 1—figure 4

At $p = 0$, $\sum_{i=1}^{25} \underline{u}_i \pi_C(i) = \sum_{i=1}^{25} \binom{25-1}{i-1} \left(\frac{1}{m}\right)^{i-1} \left(\frac{m-1}{m}\right)^{25-i} \cdot \left(0.5 + 2\left(\frac{i}{25}\right)^{0.5}\right) = 1.54546$, while

$\pi_D(0) = 1 + 2\left(\frac{0}{25}\right)^{0.5} = 1$, and therefore $\sum_{i=1}^{25} \underline{u}_i \pi_C(i) - \pi_D(0) = 0.54546$. According to the counterfactual method, benefits are

$(n-1)(\pi_D(1) - \pi_D(0)) = 24\left(1 + 2\left(\frac{1}{25}\right)^{0.5} - \left(1 + 2\left(\frac{0}{25}\right)^{0.5}\right)\right) = \frac{48}{5}$, costs are

$\pi_D(0) - \pi_C(1) = 1 + 2\left(\frac{0}{25}\right)^{0.5} - \left(0.5 + 2\left(\frac{1}{25}\right)^{0.5}\right) = \frac{1}{2} - \frac{2}{5}$ and inclusive fitness is $\frac{1}{4}\frac{48}{5} - \frac{1}{2} + \frac{2}{5} = 2.3$.

According to the regression method, as for **Figure 3**, but now with the fitness function of **Figure 4**, costs are 0.350649, benefits are 3.58443, and inclusive fitness is 0.54546.

At $p = 1$, $\pi_C(25) = 0.5 + 2\left(\frac{25}{25}\right)^{0.5} = 2.5$, while

$$\sum_{i=0}^{25-1} \bar{u}_i \pi_D(i) = \sum_{i=0}^{25-1} \binom{25-1}{i}\left(\frac{m-1}{m}\right)^i\left(\frac{1}{m}\right)^{25-i-1}\cdot\left(1+2\left(\frac{i}{25}\right)^{0.5}\right) = 2.69403,$$ and therefore

$\pi_C(25) - \sum_{i=0}^{25-1}\bar{u}_i\pi_D(i) = -0.19403$. According to the counterfactual method, benefits are

$(n-1)(\pi_C(25) - \pi_C(24)) = 24\left(0.5 + 2\left(\frac{25}{25}\right)^{0.5} - \left(0.5 + 2\left(\frac{24}{25}\right)^{0.5}\right)\right) \approx 0.9698$, costs are

$\pi_D(24) - \pi_C(25) = 1 + 2\left(\frac{24}{25}\right)^{0.5} - \left(0.5 + 2\left(\frac{25}{25}\right)^{0.5}\right) \approx 0.4596$, and inclusive fitness is

$\frac{1}{4}0.9698 - 0.4596 \approx -0.217$. According to the regression method, as for **Figure 3**, but now with the fitness function of **Figure 4**, costs are 0.45629, benefits are 1.04905, and inclusive fitness is $-0.19403$.

### Appendix 1—figure 5 (Figure 2d in the main text)
At $p = 0.473$ and $n = 25$,

$$\bar{\pi}_C(p) = \frac{1}{p}\sum_{i=0}^n \frac{i}{n}f_i(p)\pi_C(i) = \frac{1}{p}\sum_{i=0}^n \frac{i}{n}\sum_{k=0}^m \binom{m}{k}(1-p)^{m-k}p^k\binom{n}{i}\left(\frac{k}{m}\right)^i\left(\frac{m-k}{m}\right)^{n-i}\left(0.5+2\left(\frac{i}{n}\right)^{0.5}\right) = 2.04617$$

and

$$\bar{\pi}_D(p) = \frac{1}{1-p}\sum_{i=0}^n \frac{n-i}{n}f_i(p)\pi_D(i) = \frac{1}{1-p}\sum_{i=0}^n \frac{n-i}{n}\sum_{k=0}^m \binom{m}{k}(1-p)^{m-k}p^k\binom{n}{i}\left(\frac{k}{m}\right)^i\left(\frac{m-k}{m}\right)^{n-i}\left(1+2\left(\frac{i}{n}\right)^{0.5}\right) = 2.04619$$

which, with 3-digit precision in the solution variable, makes $p^* = 0.473$ the equilibrium frequency.

According to the counterfactual method, aggregate average benefits, at $p^* = 0.473$ and $n = 25$ are

$$\begin{aligned}b(p) &= \sum_{i=0}^n f_i(p)\left[\frac{i}{n}\{(i-1)(\pi_C(i)-\pi_C(i-1)) + (n-i)(\pi_D(i)-\pi_D(i-1))\}\right.\\ &\left.+\frac{n-i}{n}\{(i)(\pi_C(i+1)-\pi_C(i)) + (n-i-1)(\pi_D(i+1)-\pi_D(i))\}\right]\\ &= \sum_{i=0}^n f_i(p)\left[\frac{i}{n}\left\{(n-1)\left(2\left(\frac{i}{n}\right)^{0.5}-2\left(\frac{i-1}{n}\right)^{0.5}\right)\right\} + \frac{n-i}{n}\left\{(n-1)\left(2\left(\frac{i+1}{n}\right)^{0.5}-2\left(\frac{i}{n}\right)^{0.5}\right)\right\}\right]\\ &= 2.10255\end{aligned}$$

Average costs, similarly computed, are

$$c(p) = \sum_{i=0}^n f_i(p)\left[\frac{i}{n}(\pi_D(i-1)-\pi_C(i)) + \frac{n-i}{n}(\pi_D(i)-\pi_C(i+1))\right] = 0.412394$$

This makes inclusive fitness at $p^* = 0.473$ and $n = 25$ equal to
$rb(p^*) - c(p^*) = \frac{1}{4}(2.10255) - 0.412394 = 0.1129435$

According to the regression method, aggregate benefits as well as costs, at $p^* = 0.473$ and $n = 25$, follow from the following minimization

$$\min_{\alpha,\beta,\gamma}\sum_{i=0}^n f_i(p)\left[\frac{i}{n}\left(\pi_C(i) - \left(\alpha + \beta\left(\frac{i-1}{n-1}\right) - \gamma\right)\right)^2 + \frac{n-i}{n}\left(\pi_D(i) - \left(\alpha + \beta\left(\frac{i}{n-1}\right)\right)\right)^2\right]$$

which results in $\alpha^* = 1.43807$, $b = \beta^* = 1.71421$, and $c = \gamma^* = 0.428575$. This makes $rb - c = 0$.

Mathematica files with further computational details can be obtained from the author.

Notice that in these examples, the costs and benefits according to the regression method are uniquely defined, because – unlike the example in the main text – there is only one type of interactant here. Any specification that includes it, will therefore result in the same regression coefficients. In **Appendix 1—figures 3**, **4** and **5** it does however still misspecify the shape of the payoff function, in the sense that it replaces a non-linear function by a linear one.

## X. Continuous population structures

Also the continuous population structure we use for the examples is meant to be convenient rather than realistic. Here we use the Beta distribution with parameters $ap$ and $a(1-p)$:

$$f_p(x) = \frac{\Gamma(a)}{\Gamma(ap)\Gamma(a(1-p))} x^{ap-1}(1-x)^{a(1-p)-1}$$

This is a consistent population structure. Firstly, the frequencies of different group types, by definition of the Gamma function, integrate to one.

Also, the frequency of co-operators – which is just the first moment of the Beta distribution – is consistent with $p$.

$$\int_0^1 x f_p(x) dx = \frac{ap}{ap + a(p-1)} = p$$

Relatedness is

$$
\begin{aligned}
r(p) \;\; &= P(C|C) - P(C|D) \\
&= \frac{\int_0^1 x^2 f_p(x) dx}{p} - \frac{\int_0^1 x(1-x) f_p(x) dx}{1-p}
\end{aligned}
$$

This makes $P(C|C)$ the second moment of the Beta distribution, over $p$, while $P(C|D)$ is the first minus the second moment, over $1-p$. Relatedness therefore is

$$
\begin{aligned}
r(p) \;\; &= \frac{\frac{ap(ap+1)}{(a+1)a}}{p} - \frac{p - \frac{ap(ap+1)}{(a+1)a}}{1-p} = \frac{\frac{p(ap+1)}{a+1}}{p} - \frac{p - \frac{p(ap+1)}{a+1}}{1-p} \\
&= \frac{\frac{p(ap+1)}{a+1}}{p} - \frac{\frac{p(a+1)}{a+1} - \frac{p(ap+1)}{a+1}}{1-p} = \frac{ap+1}{a+1} - \frac{ap}{a+1} = \frac{1}{a+1}
\end{aligned}
$$

This is independent of $p$. The population structure used for **Appendix 1—figure 6** has $a = 3$, and hence relatedness $\frac{1}{4}$. The computations of $\overline{\pi}_C(p) - \overline{\pi}_D(p)$ and of inclusive fitness are done by combining this population structure with the payoff function from the example, in the way suggested in Section VIII.

