## [Decision Letter]

[Editors’ note: a previous version of this study was rejected after peer review, but the authors submitted for reconsideration. The first decision letter after peer review is shown below.]

Thank you for submitting your work entitled "How to test Hamilton's rule empirically" for consideration by *eLife*. Your article has been reviewed by a Senior Editor, a Reviewing Editor and three peer reviewers. The following individuals involved in review of your submission have agreed to reveal their identity: Burt Simon (Reviewer #1); Eörs Szathmáry (Reviewer #2).

As you can see from the reviewer reports, opinions are divided about the merits of the paper. Based on these reports, your paper is unfortunately not publishable in *eLife* in its current form. However, we encourage you to resubmit a new version of your manuscript, as we think that it contains an important and useful message.

I strongly encourage you to consider the following points in case you decide to submit a new version of the paper.

1) Two reviewers mentioned that the paper should have a different title, and one report goes into some details explaining the reasons for this. I concur with those explanations. Essentially, your paper contains a nice mathematical model that allows you to apply the counterfactual method for calculating HR and to classify situations in which HR predicts the directions of evolution, and situations in which it does not. After explaining that HR always trivially makes the right predictions when HR is calculated using the regression method, I think the model and the classification of different cases should take central stage. I concur with reviewer 1 that putting the question of empirical tests of HR in the centre misses the mark, and that the empirical perspective should be scaled back and only discussed after the model is explained. I encourage you to take the comments of reviewer 1 into consideration in this regard. Putting the experimental perspective upfront, and in particular putting it in the title, slants the paper in wrong direction in our view. (My suggestion for a title would e.g. contain something like "testing HR with a mathematical model…".)

2) One of the reviewers suggest that your results are not new. I agree with this to some extent, e.g. with regard to the tautological nature of the regression method. I do not share the reviewer's view with regard to the counterfactual model. The reviewer points to two figures in one of your 2017 papers, however, despite those figures being similar to Figure 4 in the current paper, the model used here is quite different, and more general, and I therefore think that its publication is warranted. However, I would urge you to explain in some detail what is different from previous publications, and in that regard, it would again seem advantageous to put the model at central stage (rather than the empirical question).

3) It would be good to present some results as to the overall "likelihood" that HR fails in the model presented. There are a few places where the argumentation is vague, and it would be good if it were made more quantitative. For example, the paper states that when the system moves away from (unstable) equilibria at which HR fails, it then moves into regions where HR holds again (e.g. subsection “Hamilton’s rule”). But as far as I can tell, the claim that HR holds in the regions into which the systems moves is not substantiated. Also, the first paragraph of the Discussion section seems to imply that HR always holds with continuous traits, but the logic there is not clear. E.g. it says that "Alternatively,…", but then does not go into whether HR holds or not in the case of branching, but only states that rather obvious fact that branching leads to conclusion. In fact, I would say that HR does not hold at a branching point, because at such a point HR predicts that there is no evolution (HR=0), even though in fact all (nearby) types can invade! Perhaps this should be discussed in more detail.

4) Finally, I would recommend to think about whether it would be a good idea to comment on the possibility of HR failing in other "general" models, e.g. the one by Lehmann et al., 2016 (Please note that including such material is not a requirement for a resubmission, I would simply recommend considering this.)

When resubmitting your paper, please address the above points, as well as all the points raised by the reviews whenever possible.

Reviewer #1:

I think the population model in the appendix is worthy of publication on its own. it contains enough parameters and flexibility to study the evolution of cooperation in a wide variety of idealized structured populations. The mathematics is both nontrivial and aesthetically pleasing, and there are a number of important aspects of the model that can be analyzed exactly. It's a very nice model that is (literally) tailor made for studying Hamilton's rule.

I'm not nearly as enthusiastic about some of the other aspects of the paper. To start, I don't think the title is a good description of the contents. From the title I was expecting the paper to describe an experiment designed to test Hamilton's rule that a scientist could perform with (say) microbes, and/or an informative review/critique of previous attempts to test HR empirically. The review of previous experiments in the last section of the main text assumes the reader is already familiar with those experiments. I was not familiar with them, and after reading the present paper, I still don't know how those experiments were designed or why the results did or didn't support HR.

So the paper is not really about empirical experiments for testing HR. What it seems to really be about is being specific about what it means for Hamilton's rule to fail in a mathematical model, and in which scenarios this is even possible. So, a better title might be "How to test Hamilton's rule with a mathematical model".

A scientist could (perhaps) read this paper and then design a proper experiment to test HR. If this is the intended purpose of the paper, then it has mostly succeeded. The reason the paper doesn't fully succeed (in my view) is that as nice as the mathematical model is, it may not really apply to any biological experiment that could actually be performed. In order for the mathematics to be relevant, the microbes would need to be cooperators or defectors with an unambiguous way to tell which is which, and groups would need to be carefully placed into separate containers so the desired population structure is in place. It would also need to be true that reproduction in the population is synchronized. It may be possible to do all these things. But (in my view) the experiment wouldn't really be necessary anyway. If someone can find parameter settings for the model where HR fails, where the parameters at least loosely align with a real biological system, then the case is closed: HR does not always hold. Why isn't this enough? If the author can make the (philosophical) argument that to show HR doesn't always hold, it is sufficient to consider mathematical models, I think that would be an important contribution.

My suggestion is that the paper should be rearranged so the model and its analysis (the Appendix) is center stage. It would be nice to then see an example worked out where (of course) the regression method yields {r,b,c} that is consistent with HR, but the counterfactual method yields a different {r,b,c} that is not. Perhaps parameter settings could be chosen randomly (repeatedly) and statistics gathered to get at the interesting question of how often HR holds in the model.

Reviewer #2:

This is very valuable paper. It contributes a lot of clarity to a field that traditionally has been shaped substantially by belief and/of tautology. There are stronger philosophical contributions to be drawn later - but that is another (scary) matter.

The paper can be published as it is.

Reviewer #3:

This paper is titled "How to test Hamilton's rule empirically." From this title, one might expect a series of instructions or guidelines on what to measure to test Hamilton's Rule in the real world, or at least a discussion and critique of how existing empirical tests have gone about doing this. But instead, the paper is almost entirely a disquisition on when different versions of Hamilton's rule may or may not hold at the equilibrium of a discrete, two-trait model.

There are two arguments in the paper, one for the regression definition of HR and one for the HR defined by what the author calls the "counter-factual" method. The summary of the two arguments are below:

1) Linear regression definition of HR always holds by construction, so no data is required, but the regression coefficients will depend on model specification.

2) The "counter-factual" HR always holds with additive fitness, and with positive non-additivity at the two edge equilibria, but with negative non-additivity, it may not hold at the interior equilibrium.

Before moving on to the content of these arguments (which I think is weak), I should say that both points have been both made, in a substantively identical way, in a previous paper by Van Veelen et al.,(2017). Point 1 is found in their section 4 and point 2 is essentially a rearranged version of the argument in their Sections 3 and 8. In fact, one of the figures in this manuscript, Figure 4, is essentially a replication of Figure 30 and Figure 31 in Van Veelen et al. As far as I can see, the only technically new addition here might be an extension of the argument for non-additive games to N-players but that is a relatively minor technical extension (the mathematical machinery of this is also present in Van Veelen et al.,). I would therefore say this manuscript essentially reproduces previously published material, with no new points being made. Thus, on novelty grounds I don't see why this is publishable in *eLife*.

1) It is trivially true that the model specification will of course change the regression coefficients inferred from a given statistical model. Does that mean any statistical fitting exercise is just futile? No, because the model fit (or the likelihood of data) given different specifications is informative about potential causal effects. This is of course the purpose of the time-honored theory of quantitative genetics which the author makes not even a passing reference to (e.g., reading McGlothlin et al., 2014, Phil Trans can give the author some idea of how one might use regression and allied statistical methods productively).

But the author isn't even technically correct when he asserts HR regression cannot fail: *of course* it can: in the sense that RB-C can be positive, but the trait might be going down in frequency due to environmental effects, correlated selection and genetic constraints, etc. The world is not identical to our models which is why we go out and test then.

2) The "failure" of the counterfactual method (at the mixed equilibrium of a game with negative synergy) essentially stems from trying to pretend non-additive model can be used to describe an additive system. So, this is not a surprise here (quite apart from the fact that the same author has already made the point), and I think this simple fact is much less useful than the author seems to think. I do believe that empirical researchers tend to use too simplistic versions of HR without thinking carefully about the structure of the interaction in their model system (e.g., see Akcay and Van Cleve, 2016), but there is nothing in the current treatment that provides any positive insight beyond essentially a rehashing of the non-additivity issue in a vaguely hostile way.

In short, I don't see why this manuscript adds to the literature on empirical testing, certainly beyond other recent papers by the same author. Therefore, I cannot recommend publication.

---

## [Author Response]

[Editors’ note: The authors submitted a new version of their paper for consideration and what now follows is the authors’ response to the first round of peer review.]

As you can see from the reviewer reports, opinions are divided about the merits of the paper. Based on these reports, your paper is unfortunately not publishable in eLife in its current form. However, we encourage you to resubmit a new version of your manuscript, as we think that it contains an important and useful message.I strongly encourage you to consider the following points in case you decide to submit a new version of the paper.1) Two reviewers mentioned that the paper should have a different title, and one report goes into some details explaining the reasons for this. I concur with those explanations. Essentially, your paper contains a nice mathematical model that allows you to apply the counterfactual method for calculating HR and to classify situations in which HR predicts the directions of evolution, and situations in which it does not. After explaining that HR always trivially makes the right predictions when HR is calculated using the regression method, I think the model and the classification of different cases should take central stage. I concur with reviewer 1 that putting the question of empirical tests of HR in the centre misses the mark, and that the empirical perspective should be scaled back and only discussed after the model is explained. I encourage you to take the comments of reviewer 1 into consideration in this regard. Putting the experimental perspective upfront, and in particular putting it in the title, slants the paper in wrong direction in our view. (My suggestion for a title would e.g. contain something like "testing HR with a mathematical model…".)

**Title.** All reviewers indeed found the old title not representative of the content. Reviewer 1 moreover prefers the focus of the paper to shift away from empirical testing, and towards the implications of the population model, and the new title should reflect that change. Reviewer 2 also suggests a title in question form that is a better teaser. I have changed it to “Can Hamilton's rule be violated?”, which I hope accommodates both wishes.

**Organization and focus of the paper.** I think I have reorganized the paper as suggested, taking the primary focus away from the empirical testing, and only discussing empirical implications at the end.

2) One of the reviewers suggest that your results are not new. I agree with this to some extent, e.g. with regard to the tautological nature of the regression method. I do not share the reviewer's view with regard to the counterfactual model. The reviewer points to two figures in one of your 2017 papers, however, despite those figures being similar to Figure 4 in the current paper, the model used here is quite different, and more general, and I therefore think that its publication is warranted. However, I would urge you to explain in some detail what is different from previous publications, and in that regard, it would again seem advantageous to put the model at central stage (rather than the empirical question).

It is true that there is overlap, and I am very happy that the Editor and reviewers 1 and 2 recognized the added value and elegance of the more general model. I also hope that it is clear that in the old version, I made no effort to hide the overlap, as I referred in detail to different sections in the 2017 paper, but I totally understand that it is helpful to indicate better what is new and what is not, and also to make the (new) general model more central. Pointing to the overlap for Figure 3 also allowed me to point to a mistake in our 2017 paper (which was entirely my mistake, and not of my co-authors).

3) It would be good to present some results as to the overall "likelihood" that HR fails in the model presented.

I agree that this would be interesting, but I don’t really know what to assume as a prior for the distributions of fitness functions and population structures. Also, I imagine that this could be such a broad question, that it would warrant a whole paper on its own. I therefore hope it is OK that I leave this question unanswered here.

There are a few places where the argumentation is vague, and it would be good if it were made more quantitative. For example, the paper states that when the system moves away from (unstable) equilibria at which HR fails, it then moves into regions where HR holds again (e.g. subsection “Hamilton’s rule”). But as far as I can tell, the claim that HR holds in the regions into which the systems moves is not substantiated.

This point is totally correct. The whole reason to focus on the extremes, where cooperators invade defectors or vice versa, was that in the interior, things are much more complicated. Therefore, I was wrong to suggest that the conditions that prevent violations at 𝑝 = 0 and 𝑝 = 1 also preclude violations in the interior. The new subsection “Not observing violations in equilibrium, also in the interior” of the Appendix hopefully solves this, by exploring possible claims about the absence of in-equilibrium violations in the interior much more precisely.

This has also led to a new version of a paragraph in the main text (subsection “Hamilton’s rule”).

Besides this, I now elaborate on the computations that went into Figure 2 of the main text (section IX "Discrete population structures" of the Appendix), including new computations of inclusive fitness for the regression method, as suggested by reviewer #1. I am also happy to make the Mathematica files available, so that readers can play around with the computations for themselves.

Also, the first paragraph of the Discussion section seems to imply that HR always holds with continuous traits, but the logic there is not clear. E.g. it says that "Alternatively,…", but then does not go into whether HR holds or not in the case of branching, but only states that rather obvious fact that branching leads to conclusion. In fact, I would say that HR does not hold at a branching point, because at such a point HR predicts that there is no evolution (HR=0), even though in fact all (nearby) types can invade! Perhaps this should be discussed in more detail.

I have rephrased the description there, which now is in subsection “Implications for empirical tests of Hamilton’s rule”. The mismatch between the prediction of HR and what happens at a branching point is now mentioned separately. I hope though that, besides those smaller changes, I can keep this discussion of continuous traits short, in hope to seduce the readers to read Section 6 of the 2017 paper, which is more precise, because it is more extensive.

4) Finally, I would recommend to think about whether it would be a good idea to comment on the possibility of HR failing in other "general" models, e.g. the one by Lehmann et al., 2016 (Please note that including such material is not a requirement for a resubmission, I would simply recommend considering this.)

I have chosen to not also consider other general models, to keep this paper from growing too much in size.

When resubmitting your paper, please address the above points, as well as all the points raised by the reviews whenever possible.Reviewer #1:I think the population model in the appendix is worthy of publication on its own. it contains enough parameters and flexibility to study the evolution of cooperation in a wide variety of idealized structured populations. The mathematics is both nontrivial and aesthetically pleasing, and there are a number of important aspects of the model that can be analyzed exactly. It's a very nice model that is (literally) tailor made for studying Hamilton's rule.

I am very glad that the model is appreciated! I am also very glad that the reviewer likes the Appendix, and I hope that, even more than before, I can seduce the readers in the main text to also read the details in the Appendix, which I hold dear.

I'm not nearly as enthusiastic about some of the other aspects of the paper. To start, I don't think the title is a good description of the contents. From the title I was expecting the paper to describe an experiment designed to test Hamilton's rule that a scientist could perform with (say) microbes, and/or an informative review/critique of previous attempts to test HR empirically. The review of previous experiments in the last section of the main text assumes the reader is already familiar with those experiments. I was not familiar with them, and after reading the present paper, I still don't know how those experiments were designed or why the results did or didn't support HR.So the paper is not really about empirical experiments for testing HR. What it seems to really be about is being specific about what it means for Hamilton's rule to fail in a mathematical model, and in which scenarios this is even possible. So, a better title might be "How to test Hamilton's rule with a mathematical model".

**Title.** I have changed the title in response to these remarks, as well as the remarks of reviewers #2 and #3 and the editor.

**Description of the empirical literature.** I hope I also improved the parts that describe the current empirical literature. Also, the reorganization – putting it after the theory part – might help already. I hope I can be excused not to go into all the details of the current empirical tests – besides Smith et al., (2010) – and focus only on the fact that none of these other papers use non-linear statistics to analyze their data, which is a necessity if the statistics is to be able to capture meaningful violations at all.

A scientist could (perhaps) read this paper and then design a proper experiment to test HR. If this is the intended purpose of the paper, then it has mostly succeeded.

This is indeed the purpose; I hope that this paper would force anyone that sets out to test Hamilton’s rule to reflect on what violations look like, where not to expect them, and to treat the data so that they can capture violations properly – that is, by allowing for non-linearities in their statistical models.

The reason the paper doesn't fully succeed (in my view) is that as nice as the mathematical model is, it may not really apply to any biological experiment that could actually be performed. In order for the mathematics to be relevant, the microbes would need to be cooperators or defectors with an unambiguous way to tell which is which, and groups would need to be carefully placed into separate containers so the desired population structure is in place. It would also need to be true that reproduction in the population is synchronized. It may be possible to do all these things.

I agree that it will be very difficult to get observations, both of a fitness function, and of a population structure. Papers like Smith et al. (2010) and Chuang et al., (2009, 2010) are examples that good empiricists can achieve at least the measuring of the fitness as a function of (local) shares of co-operators, as the reviewer suggests, but it is true that more remains to be done, including capturing the population structure.

But (in my view) the experiment wouldn't really be necessary anyway. If someone can find parameter settings for the model where HR fails, where the parameters at least loosely align with a real biological system, then the case is closed: HR does not always hold. Why isn't this enough? If the author can make the (philosophical) argument that to show HR doesn't always hold, it is sufficient to consider mathematical models, I think that would be an important contribution.

I very much agree that it is already interesting in and of itself to see how Hamilton’s rule, in theory, can be violated. I do however think that in this case it is also worth pointing to the remarkable state of the empirical literature, which is mostly guided by what I think is a naïve idea of what violations of Hamilton’s rule would look like. If theory can finally make a link to help do better empirics, I think it is worth discussing empirical implications. I hope that it is OK that, although the main focus of the paper is now on the general model itself, it still ends with implications for empirical tests.

My suggestion is that the paper should be rearranged so the model and its analysis (the Appendix) is center stage. It would be nice to then see an example worked out where (of course) the regression method yields {r,b,c} that is consistent with HR, but the counterfactual method yields a different {r,b,c} that is not. Perhaps parameter settings could be chosen randomly (repeatedly) and statistics gathered to get at the interesting question of how often HR holds in the model.

I have rearranged the paper, also following similar suggestions by the editor. In the Appendix, I now elaborate (section IX) on the computations that went into Figure 2 of the main text, and, in response to this comment, I have also added computations of costs and benefits, and thereby inclusive fitness, according the regression method, for the different panels of Figure 2 (which are mostly also separate figures in the Appendix).

Reviewer #2:This is very valuable paper. It contributes a lot of clarity to a field that traditionally has been shaped substantially by belief and/of tautology. There are stronger philosophical contributions to be drawn later - but that is another (scary) matter.The paper can be published as it is.

I am very glad to hear this!

Reviewer #3:This paper is titled "How to test Hamilton's rule empirically." From this title, one might expect a series of instructions or guidelines on what to measure to test Hamilton's Rule in the real world, or at least a discussion and critique of how existing empirical tests have gone about doing this. But instead, the paper is almost entirely a disquisition on when different versions of Hamilton's rule may or may not hold at the equilibrium of a discrete, two-trait model.

Since all reviewers found that the title did not match the content well enough, it has been changed.

There are two arguments in the paper, one for the regression definition of HR and one for the HR defined by what the author calls the "counter-factual" method. The summary of the two arguments are below:1) Linear regression definition of HR always holds by construction, so no data is required, but the regression coefficients will depend on model specification.2) The "counter-factual" HR always holds with additive fitness, and with positive non-additivity at the two edge equilibria, but with negative non-additivity, it may not hold at the interior equilibrium.

This is an almost-complete summary of the results. There are two possible elements of increased precision, both at point 2.

a) There is a difference between HR not holding – which in fact happens at 𝑝 = 0 and 𝑝 = 1 for any deviation of additivity – and not holding in equilibrium – which in some cases (see Figure 3A) can indeed not happen at the corners.

b) For other cases, it can actually even in equilibrium be violated at 𝑝 = 0 and 𝑝 = 1 (see Figure 3B). This is also a point where our JTB paper was incorrect, so I of course cannot blame the reviewer for overlooking this possibility here too. I now point out that our 2017 paper did not draw the right conclusion in subsection “Hamilton’s rule” of the manuscript (this mistake was entirely mine, and not of my co-authors).

Before moving on to the content of these arguments (which I think is weak), I should say that both points have been both made, in a substantively identical way, in a previous paper by Van Veelen et al., (2017). Point 1 is found in their section 4 and point 2 is essentially a rearranged version of the argument in their Sections 3 and 8. In fact, one of the figures in this manuscript, Figure 4, is essentially a replication of Figure 30 and Figure 31 in Van Veelen et al. As far as I can see, the only technically new addition here might be an extension of the argument for non-additive games to N-players but that is a relatively minor technical extension (the mathematical machinery of this is also present in Van Veelen et al.,). I would therefore say this manuscript essentially reproduces previously published material, with no new points being made. Thus, on novelty grounds I don't see why this is publishable in eLife.

I agree that some of the points are made previously in our 2017 paper. (Just to be sure: I think I have also made no effort to hide the overlap, as I referred, also before, in detail to different sections in the 2017 paper). I do however disagree that there is not enough value added – although I see that this is perhaps a matter of taste. The new model I think is much more general, and certainly not all of the mathematical machinery is already present in the 2017 paper; for instance, the quite central idea of a population structure profile is entirely absent in the 2017 paper.

1) It is trivially true that the model specification will of course change the regression coefficients inferred from a given statistical model. Does that mean any statistical fitting exercise is just futile? No, because the model fit (or the likelihood of data) given different specifications is informative about potential causal effects. This is of course the purpose of the time-honored theory of quantitative genetics which the author makes not even a passing reference to (e.g., reading McGlothlin et al., 2014, Phil Trans can give the author some idea of how one might use regression and allied statistical methods productively).

This remark does not match with what it is that the paper claims, and ‘refutes’ things that I am not saying.

1) I do not claim that it is deep or non-trivial that regression coefficients change with model specifications. Quite the contrary: I hope that the familiarity of those that do statistics with this straightforward observation, will actually help bring the point across. All I claim is that estimators of regression coefficients vary with specification. Nothing more, nothing less. This is the core ingredient for observing that costs and benefits in Hamilton’s rule are not necessarily uniquely defined, when using the regression method. The paper I think was and is quite clear that this (Hamilton’s rule not being uniquely defined with the regression method) is the point.

2) I do not claim that statistical fitting exercises are futile. There is really nothing in the text to suggest that. What I would say actually goes in the opposite direction. For making a good fit, it is not productive to do the fitting exercise within an a priori restricted set of linear models only – which is what the regression method does – especially when the data would suggest otherwise. And for fit, it is an even worse (actually an absurd) idea in modelling exercises, in which the postulated model is explicitly not linear, and you could get perfect fit by just taking the postulated model itself, and not replacing it with something linear. The linear model then produces Hamilton’s rule, according to the regression method, but that goes at the expense of the fit, in a case where we moreover *know for sure* there is a mismatch. So in fact I care very much – and not very little – for good fit!

But the author isn't even technically correct when he asserts HR regression cannot fail: *of course* it can: in the sense that RB-C can be positive, but the trait might be going down in frequency due to environmental effects, correlated selection and genetic constraints, etc. The world is not identical to our models which is why we go out and test then.

Again, this is a comment that disagrees with something I am not claiming, and that overlooks what it is that I am claiming.

Here it will be helpful to separate “models of how the world works” on the one hand from Hamilton’s rule on the other. Hamilton’s rule is not a model. It is a rule, that one can define (in different ways, depending on the definition of costs and benefits) for any given model. The generality of Hamilton’s rule with the regression method is that, whichever model of the world one chooses, Hamilton’s rule will always agree with the direction of selection. So, while any given model of the world could (indeed, of course) be rejected, the fact that for any conceivable way in which the world could work (including the way the world really works), Hamilton’s rule, with costs and benefits according to the regression method, holds, means that this version of Hamilton’s rule cannot fail.

I have rephrased one small part of the paper to stress this point a bit more (now subsection “Implications for empirical tests of Hamilton’s rule”).

2) The "failure" of the counterfactual method (at the mixed equilibrium of a game with negative synergy) essentially stems from trying to pretend non-additive model can be used to describe an additive system. So, this is not a surprise here (quite apart from the fact that the same author has already made the point), and I think this simple fact is much less useful than the author seems to think. I do believe that empirical researchers tend to use too simplistic versions of HR without thinking carefully about the structure of the interaction in their model system (e.g., see Akcay and Van Cleve, 2016), but there is nothing in the current treatment that provides any positive insight beyond essentially a rehashing of the non-additivity issue in a vaguely hostile way.

This remark is upside down in an almost dazzling way. It is not the counterfactual method that, as the reviewer writes, is “trying to pretend a non-additive model can be used to describe an additive system”. That would be weird, because additive models are nested in the larger set of non-additive ones, so if that were the case – which it is not, because the counterfactual method just is not doing that – then that could never even be a problem. The point that I make, is the double contrary of what the reviewer writes. It is the regression method that is trying to describe a non-additive system with an additive model. That is where a real difference between the two methods can arise, since non-additive models are *not* a subset of additive ones. Again, it is remarkable that the reviewer reads this into the paper, because both the theory and the examples all clearly point towards the issues at hand arising when the real world is not additive, and the different methods for computing costs and benefits diverge, and not when the real world is additive, in which case both methods (with the “right” specification for the regression method) just find the same costs and benefits, and Hamilton’s rule for both agrees with the direction of selection.

The fact that the reviewer seems to shrug his or her shoulders about empirical researchers tending to use simplistic versions without thinking carefully about the structure of the interaction in their model, I also find strange. I think it would be of obvious importance for empirical researchers to think carefully about this, because for explicit tests of Hamilton’s rule, this makes the difference between a test that is meaningless, because it by design does not allow for non-spurious violations, and a test that is meaningful, because it can, if the real world happens to be non-linear, in principle reject Hamilton’s rule.

Finally, I think the paper is not vaguely hostile at all. I do not really know how to respond to this accusation, but I do find it very unpleasant that bringing up factual points about limitations of inclusive fitness, here and in other instances, apparently cannot be done without being accused of hostility, in spite of the fact that I appreciate the value of Hamilton’s rule very much, and have expressed that appreciation regularly.

In short, I don't see why this manuscript adds to the literature on empirical testing, certainly beyond other recent papers by the same author. Therefore, I cannot recommend publication.

The (sometimes creative) mis-reading of the manuscript, and the fact that this happens more often, to me indicates that there is a relatively general unwillingness to seriously consider the possibility that there can be limitations of Hamilton’s rule. Although being defensive about something as valuable as Hamilton’s rule is understandable, this unwillingness is not rooted in scientific considerations only, and therefore this report to me only underscores the importance of publishing a balanced demarcation of when Hamilton’s rule can, and when it cannot be violated.